# Risk-Monotonicity in Statistical Learning

**Zakaria Mhammedi**[*]
Massachusetts Institute of Technology
mhammedi@mit.edu

## Abstract

Acquisition of data is a difficult task in many applications of machine learning, and it is only natural that one hopes and expects the population risk to decrease (better performance) *monotonically* with increasing data points. It turns out, somewhat surprisingly, that this is not the case even for the most standard algorithms that minimize the empirical risk. Non-monotonic behavior of the risk and instability in training have manifested and appeared in the popular deep learning paradigm under the description of *double descent*. These problems highlight the current lack of understanding of learning algorithms and generalization. It is, therefore, crucial to pursue this concern and provide a characterization of such behavior. In this paper, we derive the first consistent and risk-monotonic (in high probability) algorithms for a general statistical learning setting under weak assumptions, consequently answering some questions posed by [53] on how to avoid non-monotonic behavior of risk curves. We further show that risk monotonicity need not necessarily come at the price of worse excess risk rates. To achieve this, we derive new empirical Bernstein-like concentration inequalities of independent interest that hold for certain non-i.i.d. processes such as Martingale Difference Sequences.

## 1 Introduction

Guarantees on the performance of machine learning algorithms are desirable, especially given the widespread deployment. A traditional performance guarantee often takes the form of a generalization bound, where the *expected* risk associated with hypotheses returned by an algorithm is bounded in terms of the corresponding empirical risk plus an additive error which typically converges to zero as the sample size increases. However, interpreting such bounds is not always straight forward and can be somewhat ambiguous. In particular, given that the error term in these bounds goes to zero, it is tempting to conclude that more data would monotonically decrease the expected risk of an algorithm such as the Empirical Risk Minimizer (ERM). However, this is not always the case; for example, [33] showed that increasing the sample size by one, can sometimes make the test performance worse in expectation for commonly used algorithms such as ERM in popular settings including linear regression. This type of non-monotonic behavior is still poorly understood and indeed not a desirable feature of an algorithm since it is expensive to acquire more data in many applications.

Non-monotonic behavior of *risk curves* [47]—the curve of the expected risk as a function of the sample size—has been observed in many previous works [18, 43, 48, 20] (see also [33, 52] for nice accounts of the literature). At least two phenomena have been identified as being the cause behind such behavior. The first one, coined *peaking* [30, 19], or *double descent* according to more recent literature [6, 49, 7, 17, 15, 37, 40, 41, 16, 12, 11, 14, 42], is the phenomenon where the risk curve peaks at a certain sample size $n$. This sample size typically represents the cross-over point from an over-parameterized to under-parameterized model. For example, when the number of data points is less than the number of parameters of a model (over-parameterized model), such as Neural

---

[*]Work done while at the Australian National University.

Networks, the expected risk can typically increase until the number of data points exceeds the number of parameters (under-parameterized model). The second phenomenon is known as *dipping* [32, 31], where the risk curve reaches a minimum at a certain sample size $n$ and increases after that—never reaching the minimum again even for very large $n$. This phenomenon typically happens when the algorithm is trained on a surrogate loss that differs from the one used to evaluate the risk [8].

It is becoming more apparent that the two phenomena just mentioned (double descent and dipping) do not fully characterize when non-monotonic risk behavior occurs [34]. [33] showed that non-monotonic risk behavior could happen outside these settings and formally prove that the risk curve of ERM is non-monotonic in linear regression with prevalent losses. The most striking aspect of their findings is that the risk curves in some of the cases they study can display a perpetual "oscillating" behavior; there is no sample size beyond which the risk curve becomes monotone—see Figure 1. In such cases, the risk's non-monotonicity cannot be attributed to the peaking/double descent phenomenon. Moreover, they rule out the dipping phenomenon by studying the ERM on the actual loss (not a surrogate loss).

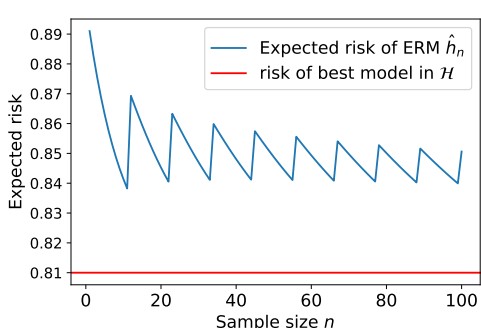

Figure 1: Expected risk of ERM on a 1d linear regression problem with absolute loss and two instances $z_1 = (x_1, y_1) = (1, 1)$ and $z_2 = (x_2, y_2) = (1/10, 1)$ such that $\mathbf{P}[Z = z_1] = 0.1$ and $\mathbf{P}[Z = z_2] = 0.9$. The set of hypotheses is the real line, i.e. $\mathcal{H} = \mathbb{R}$. The ERM solution $\hat{h}_n$ admits a closed form in this case—see [33] for details.

The findings of [33] stress our current lack of understanding of generalization. This was echoed more particularly by [53], who posed the following question as part of a COLT open problem:

*How can we provably avoid non-monotonic behavior?*

While excess risk bounds are typically monotonic, this does not guarantee the monotonicity of the actual risk. In this work, we study under which assumptions on the learning problem there exist consistent and risk monotonic algorithms. We also aim to quantify the price to pay, in terms of corresponding excess risk rates, for achieving risk monotonicity.

**Contributions.** In this work, we answer some questions posed by [53] by presenting an algorithm that is both consistent and risk-monotonic in high probability under weak assumptions on the learning problem. Our algorithm is technically a "wrapper" that takes as input any base learning algorithm B and makes up a new algorithm A that is risk monotonic in high probability and enjoys essentially the same excess risk rate as B. Crucially, our results show that risk monotonicity need not come at the expense of worse excess risk rates. In fact, we show that fast rates are achievable under a Bernstein condition (Definition 3).

Our results hold under the general statistical learning setting with a bounded loss. We even go beyond the standard i.i.d. assumption on the loss process. Our relaxed technical condition on the loss process, which is formalized in Assumption 1 below, is reminiscent of the condition characterizing Martingale Difference Sequences (MDS). In a nutshell, we will assume a setting where the instance random variables $Z_1, Z_2, \ldots$ and the loss $\ell$ satisfy, for all hypotheses $h$, $\mathbf{E}[\ell(h, Z_t) \mid Z_1, \ldots, Z_{t-1}] = L(h)$, for some risk function $L$. This is trivially satisfied in the i.i.d. case, where $L$ corresponds to the standard risk function. In general, this condition may be satisfied even if $Z_1, Z_2, \ldots$ are dependent or have different marginal distributions. We argue that our relaxed assumption on the loss process is the weakest assumption under which studying risk-monotonicity still makes sense.

To achieve risk monotonicity under our loss process assumption, we derive a new concentration inequality/generalization bound of PAC-Bayesian flavor for MDS (see Proposition 5). This concentration inequality may be thought of as an empirical Freedman's inequality [23] or as an extension of the empirical Bernstein inequality [35] to MDS. Our concentration inequalities also have the advantage of being time-uniform with the optimal dependence on the number of samples. Here, time-uniform means that the inequalities hold for all sample sizes simultaneously. While standard concentration inequalities can be turned into time-uniform ones using a union bound over the number of samples

$n$, the resulting bounds will have a sub-optimal $\ln n$ factor instead of the optimal[2] $\ln \ln n$ that we are able to get. Finally, our concentration bounds are easily derived using the guarantee of a recent parameter-free online learning algorithm—FREEGRAD[39]. Our approach opens up the door for obtaining new concentration inequalities through the design of online learning algorithms.

**Approach Overview.** Our approach to deriving the new concentration inequalities is based on the guarantee of the recent FREEGRAD algorithm. The algorithm operates in rounds, where at each round $t$, FREEGRAD outputs $\widehat{w}_t$ in some convex set $\mathcal{W}$, say $\mathbb{R}^d$, then observes a vector $\boldsymbol{g}_t \in \mathbb{R}^d$, typically the sub-gradient of a loss function at the iterate $\widehat{w}_t$. The algorithm guarantees a regret bound of the form $\sum_{t=1}^T \boldsymbol{g}_t^\top (\widehat{w}_t - \boldsymbol{w}) \leq \widetilde{O}(\|\boldsymbol{w}\| \sqrt{Q_T})$, for all $\boldsymbol{w} \in \mathcal{W}$, where $Q_T := \sum_{t=1}^T \|\boldsymbol{g}_t\|^2$. What is more, FREEGRAD's outputs $(\widehat{w}_t)$ ensure the following (see [39, Theorem 5]):

$$\widehat{w}_t^\top \boldsymbol{g}_t + \Phi(S_t, Q_t) \leq \Phi(S_{t-1}, Q_{t-1}), \quad \forall t \geq 1, \tag{1}$$

where $S_t := \|\sum_{i=1}^t \boldsymbol{g}_i\|$, $Q_t := \sum_{i=1}^t \|\boldsymbol{g}_i\|^2$, and $\Phi(S, V) := \exp(\frac{S^2/2}{\gamma^2 + V + |S|} - \frac{1}{2} \ln \frac{\gamma^2}{\gamma^2 + V})$, for any $\gamma > 0$. Instantiating this guarantee in 1d with $(\boldsymbol{g}_t)$ set to an MDS $(X_t)$ and taking (conditional) expectation in (1) shows that $\Phi_t := \Phi(\sum_{i=1}^t X_i, \sum_{i=1}^t X_i^2)$ is a non-negative supermartingale, from which concentration results can be obtained via Ville's inequality (a generalization of Markov's inequality—see Lemma 18). Our proof technique is similar to the one introduced in [28], with the difference that we use the specific shape of FREEGRAD's potential function to build our supermartingale, which leads to a desirable empirical variance term in the final concentration bound.

On the side of risk monotonicity, given $n$ samples, the key idea behind our approach is to iteratively generate a sequence of distributions $P_1, P_2, \ldots$ leading up to $P_n$ over hypotheses, where we only allow consecutive distributions, say $P_{k-1}$ and $P_k$ to differ if we can guarantee (with high enough confidence) that the risk associated with $P_k$ is lower than that of $P_{k-1}$. To test for this, we compare the average empirical losses of hypotheses sampled from $P_{k-1}$ versus ones sampled from $P_k$, taking into account the potential gap between empirical and population expectations. Applying our new concentration bounds to quantify this gap not only allows us to achieve risk monotonicity under a non-i.i.d. loss process but also enables us to achieve fast excess risk rates under the Bernstein condition. For the latter, it was crucial to have an empirical loss variance term in the concentration inequality.

**Related Works.** Much work has already been done in efforts to mitigate the non-monotonic behavior of risk curves [54, 42, 33]. For example, in the supervised learning setting with the zero-one loss, [9] introduced the "memorize" algorithm that predicts the majority label on any test instance $x$ that was observed during training; otherwise, a default label is predicted. [9] showed that this algorithm is risk-monotonic. However, it is unclear how their result could generalize beyond the particular setting they considered. Risk-monotonic algorithms are also known for the case where the model is correctly specified (see [33] for an overview); in this paper, we do not make such an assumption.

Closer to our work is that of [54] who, like us, also used the idea of only updating the current predictor for sample size $n$ if it has a lower risk than the predictor for sample size $n-1$. They determine whether this is the case by performing statistical tests on a validation set (or through cross-validation). They introduce algorithm wrappers that ensure that the risk curves of the final algorithms are monotonic with high probability. However, their results are specialized to the 0-1 loss and they do not answer the question by [53] on the existence of learners that guarantee a monotonic risk in *expectation*.

On the side of concentration bounds for non-i.i.d. processes, our results are somewhat similar to those found in e.g. [27, 26]. However, our technique for deriving them, which relies on the guarantee of a parameter-free online learning algorithm, is entirely different. The theoretical link between online regret and concentration inequalities was previously drawn—see e.g. [45, 22]. However, our approach is slightly different as we use the monotonicity of an online algorithm's potential function to get our concentration results. Thus, new concentration inequalities may be derived similarly by modifying the explicit potential function directly. Our approach is more similar to that of [28] who also derived concentration inequalities using guarantees of online betting algorithms that bet fractions smaller than one of their wealth at each round.

---

[2]One can not improve on the $\ln \ln n$ factor by the law of iterated logarithm [13].

We allow for a parameter $h \in \mathcal{H}$ in our concentration bounds to make the results useful in the statistical learning setting. These bounds are of PAC-Bayesian type and are somewhat reminiscent of those in [50, 38]. We refer the reader to [25] for an overview of existing PAC-Bayesian bounds.

**Outline.** In Section 2, we introduce the setting, notation, and relevant definitions. In Section 3, we present our new concentration inequalities for Martingale Difference Sequences and loss processes we are interested in (those that satisfy Assumption 1 below). In Section 4, we present our risk-monotonic algorithm wrapper and show that it achieves risk monotonicity in high probability. We conclude with a discussion in Section 5. The proofs of the new concentration inequalities and risk monotonicity are differed to Appendices B and C, respectively. Appendix A presents existing and new technical results needed in some of our proofs.

## 2   Preliminaries

In this section, we present the setting, notation, and relevant definitions for the rest of the paper.

**Setting and Notation.** Throughout, we will assume an underlying probability space $(\Omega, \mathcal{F}, \mathbf{P})$. Let $\mathcal{Z}$ [resp. $\mathcal{H}$] be an arbitrary feature [resp. hypothesis] space, and let $\ell : \mathcal{H} \times \mathcal{Z} \to [0, 1]$ be a bounded loss function. We denote by $\triangle(\mathcal{H})$ the set of probability measures on $\mathcal{H}$. Data is represented by random variables $Z_1, Z_2, \dots \in \mathcal{Z}$ that we assume are accessible to a learning algorithm in a sequential fashion. We use the concise notation $Z_{1:t}$ for the tuple $(Z_1, \dots, Z_t)$ and denote by $\mathcal{G}_t$ the $\sigma$-algebra generated by the random variables $Z_1, \dots, Z_t$, with the convention that $\mathcal{G}_0 = \varnothing$. We will write $\mathbf{E}_{t-1}[\cdot] \coloneqq \mathbf{E}[\cdot \mid \mathcal{G}_{t-1}]$, for $t \geq 1$.

We will not assume that the random variables $(Z_t)$ are independent and identically distributed. Instead, we will make the following weaker assumption on the loss process $(\ell(h, Z_t))$:

**Assumption 1** (Process Assumption). *There exists a risk function $L : \mathcal{H} \to [0, 1]$ such that the sequence of random variables $Z_1, Z_2, \dots$ satisfy $\mathbf{E}_{t-1}[\ell(h, Z_t)] = L(h)$, for all $h \in \mathcal{H}$ and $t \geq 1$.*

When the random variables $Z_1, Z_2, \dots$ are i.i.d. and $Z_i \sim P_Z$, $i \in \mathbb{N}$, then Assumption 1 trivially holds with the standard risk function $L(h) = \mathbf{E}_{P_Z(z)}[\ell(h, z)]$. In Assumption 1, the conditional distribution of $Z_t$ given $\mathcal{G}_{t-1}$ may be arbitrary as long as the corresponding conditional expectation $\mathbf{E}_{t-1}$ of the loss of a given hypothesis $h$ is the same (equal to $L(h)$) for all $t \geq 1$. Arguably, Assumption 1 represents the weakest condition under which studying risk monotonicity in the statistical learning setting still makes sense. We touch more on this point after defining risk monotonicity below. To simplify notation for the rest of this paper, we let

$$L(Q) \coloneqq \mathbf{E}_{Q(h)}[L(h)], \quad \text{for all } Q \in \triangle(\mathcal{H}),$$

where $L$ is as in Assumption 1.

A learning algorithm $\mathsf{A}$ is a map from $\bigcup_{i=1}^{\infty} \mathcal{Z}^i$ to $\triangle(\mathcal{H})$; given data $Z_{1:t}$, the output $\mathsf{A}(Z_{1:t})$ of the algorithm is a distribution over hypotheses in $\mathcal{H}$. This definition includes deterministic algorithms for which the distribution $\mathsf{A}(Z_{1:t})$ is a Dirac at some $h = h(Z_{1:t})$. We will use the notation $\mathsf{A}(\cdot \mid Z_{1:t}) \coloneqq \mathsf{A}(Z_{1:t})(\cdot)$.

Throughout, we will make use of a fixed "prior" distribution $P_0$ over hypotheses in $\mathcal{H}$. In Section 4, we will present an algorithm wrapper that takes any base algorithm $\mathsf{B}$ as input and makes a risk-monotonic algorithm out of it with essentially the same excess risk rate. The results of this paper are useful for base algorithms that output distributions that are absolutely continuous w.r.t. our choice of prior $P_0$; that is $\mathsf{B}(Z_{1:t}) \ll P_0$, for all $t \geq 1$[3]. In practice, if $\mathcal{H} = \mathbb{R}^d$, $P_0$ may be a multivariate Gaussian around the origin and $\mathsf{B}(Z_{1:t})$ may also be a multivariate Gaussian around the ERM $\hat{h}_n \in \arg\inf_{h \in \mathcal{H}} \sum_{i=1}^n \ell(h, Z_i)$, in which case $\mathsf{B}(Z_{1:t}) \ll P_0$ holds for all $t \geq 1$. We now define the notion of risk monotonicity we will work with:

**Definition 1** (Risk Monotonicity). *For $\delta \in (0, 1)$ and $N \geq 1$, we say that a learning algorithm $\mathsf{A} : \bigcup_{i=1}^{\infty} \mathcal{Z}^i \to \triangle(\mathcal{H})$ is $(\delta, N)$-risk-monotonic if, with probability at least $1 - \delta$,*

$$\forall t \geq N, \quad \mathbf{E}_{A(h|Z_{1:t})}\left[\mathbf{E}_t[\ell(h, Z_{t+1})]\right] \leq \mathbf{E}_{A(h|Z_{1:t-1})}\left[\mathbf{E}_{t-1}[\ell(h, Z_t)]\right]. \tag{2}$$

---

[3]We inherit this restriction from the PAC-Bayesian approach that we use to quantify generalization. In Appendix D, we show how this restriction can be removed in the i.i.d. setting (see also Remark 7).

Note that since the loss $\ell$ is positive, Fubini's theorem implies that $\mathbf{E}_{A(h|Z_{1:t-1})}\left[\mathbf{E}_{t-1}[\ell(h, Z_t)]\right] = \mathbf{E}_{t-1}\left[\mathbf{E}_{A(h|Z_{1:t-1})}[\ell(h, Z_t)]\right]$, for all $t \geq 1$. Thus, the condition in (2) requires the expected loss of algorithm A on the next sample, conditioned on the past data, to decrease with the size of the data. We note that if Assumption 1 does not hold and $\mathbf{E}_t[\ell(h, Z_{t+1})]$ depends on $t$ in an arbitrary fashion, then requiring (2) would be too strong. Thus, we will restrict our attention to processes that satisfy Assumption 1. We stress that the condition in this assumption is weaker than i.i.d., and allows the random variables $Z_1, Z_2, \ldots$ to have different (conditional) distributions as long as the moment constraint $\mathbf{E}_{t-1}[\ell(h, Z_t)] = L(h)$ is satisfied for all $h \in \mathcal{H}$ and $t \geq 1$.

The notion of monotonicity presented in [33, 53] concerned only i.i.d. random variables, and requires the risk to be monotonic in expectation as opposed to in high probability. In particular, the strongest notation of monotonicity in [33], which they refer to as global $\mathcal{Z}$-monotonicity, can be expressed as

$$\forall t \geq 1, \quad \mathbf{E}\left[\mathbf{E}_{A(h|Z_{1:t})}[L(h)]\right] \leq \mathbf{E}\left[\mathbf{E}_{A(h|Z_{1:t-1})}[L(h)]\right], \tag{3}$$

where $L(h) \coloneqq \mathbf{E}_{P_Z(z)}[\ell(h, z)]$ and $Z_1, Z_2 \ldots \overset{\text{i.i.d.}}{\sim} P_Z$. We will show that achieving risk-monotonicity in expectation up to a small fast rate term is as easy as achieving $(\delta, N)$-risk-monotonicity—at least for bounded losses. In fact, we will show how our risk-monotonic (in the sense of Def. 1) algorithm can easily be turned into one that satisfies (3) up to a fast rate term.

Monotonicity alone is rather easy to achieve; it suffices to output a fixed hypothesis $h \in \mathcal{H}$ regardless of the training dataset. In this case, the risk would be constant, and so risk monotonicity is achieved by definition. In practice, it is important to generate hypotheses with low risk, and so a fixed hypothesis that does not dependent on data is likely to be useless. Formally, we want algorithms that are risk monotonic and *consistent*:

**Definition 2** (Consistency). *Under Assumption 1, we say that algorithm* A *is* consistent *if for any* $\epsilon > 0$, $\lim_{n \to \infty} \mathbf{P}\left[|\mathbf{E}_{A(h|Z_{1:n})}[L(h)] - \inf_{h \in \mathcal{H}} L(h)| > \epsilon\right] = 0$.

We will go beyond the notation of consistency and study the rate of convergence of the risk of our algorithm to the optimal risk. We do this under the assumption that the loss process $\ell(h, Z_t)$ satisfies the Bernstein condition for $h \in \mathcal{H}$:

**Definition 3** (Bernstein Condition). *For $\beta \in [0, 1]$ and $B > 0$, the $(\beta, B)$-Bernstein condition holds if the random variables $Z_1, Z_2, \ldots$ and the loss $\ell$ satisfy, for all $t \geq 1$ and all $h \in \mathcal{H}$,*

$$\mathbf{E}_{t-1}\left[(\ell(h, Z_t) - \ell(h_\star, Z_t))^2\right] \leq B\mathbf{E}_{t-1}[\ell(h, Z_t) - \ell(h_\star, Z_t)]^\beta,$$

*for $h_\star \in \arg\inf_{h \in \mathcal{H}} \mathbf{E}_{t-1}[\ell(h, Z_t)]$.*

The Bernstein condition [3, 4, 5, 21, 29] essentially characterizes the easiness of the learning problem. In particular, it implies that the conditional variance of the excess-loss random variable $\ell(h, Z_t) - \ell(h_\star, Z_t)$ vanishes when the risk associated with the hypothesis $h \in \mathcal{H}$ gets closer to the $\mathcal{H}$-optimal risk $L(h_\star)$. For bounded loss functions, the Bernstein condition with $\beta = 0$ always holds, and so the results of this paper are always true for $\beta = 0$. The Bernstein condition with $\beta = 1$ corresponds to the easiest learning setting. The case where $\beta \in (0, 1)$ interpolates naturally between these two extremes, where intermediate excess-risk rates are achievable. We refer the reader to [29, Section 3] for examples of learning settings where a Bernstein condition holds.

**Additional useful definitions.** For $\rho > 1$ and $\delta \in (0, 1)$, we define

$$c \coloneqq \sum_{k \geq 1} \frac{1}{k \ln^2(k+1)} \approx 3.2; \quad \phi_\rho(n) \coloneqq c\sqrt{\rho+1}(\ln_\rho(n)+1)\ln^2(\ln_\rho(n)+2); \tag{4}$$

$$\text{and} \quad n_\delta \coloneqq \sup\{n \in \mathbb{N} : 8\ln(\phi_\rho(n)/\delta) > n\}. \tag{5}$$

Note that $n_\delta$ is not too large as a function of $1/\delta$. In fact, the definitions of $\phi_\rho$ and $n_\delta$ imply that $n_\delta \leq O(\ln(1/\delta))$. In the next section, we present some new concentration inequalities of independent interest that will be useful in achieving risk-monotonicity under Assumption 1 while maintaining good excess risk rates.

## 3 New (PAC-Bayesian) Concentration Inequalities

In this section, we present some new concentration inequalities of PAC-Bayesian flavor that will be crucial to deriving our risk monotonic algorithm wrapper under Assumption 1. These concentration

inequalities hold for non-i.i.d. data (which we require to accommodate Assumption 1), and are so-called *time-uniform*; the inequalities hold for all sample sizes simultaneously given a fixed confidence level. We explain below the advantage that this has in our setting.

We start by a new concentration inequality for Martingale Difference Sequences, from which we derive the bound we need under Assumption 1. First, we give the formal definition of an MDS:

**Definition 4.** *Let $(\mathcal{F}_t)_{t\in\mathbb{N}}$ be a filtration w.r.t. the underlying probability space $(\Omega, \mathcal{F}, \mathbf{P})$, i.e. $(\mathcal{F}_t)_{t\in\mathbb{N}}$ is a sequence of non-decreasing sub-$\sigma$-algebras of $\mathcal{F}$. A sequence of random variables $(X_t)$ is an MDS w.r.t. $(\mathcal{F}_t)_{t\in\mathbb{N}}$, if for all $t \geq 1$, $X_t$ is $\mathcal{F}_t$-measurable; $\mathbf{E}[|X_t|] < \infty$; and $\mathbf{E}[X_t \mid \mathcal{F}_{t-1}] = 0$ a.s.*

With this in hand, we present our first concentration inequality:

**Proposition 5** (PAC-Bayes for MDS). *Let $\rho > 1$ and $\phi_\rho$ be as in (4). Further, let $\{X_t^h\}$ be a family of random variables taking values in $[-1, 1]$ and $(\mathcal{F}_t)_{t\in\mathbb{N}}$ be a filtration such that $(X_t^h)$ is an MDS w.r.t to $(\mathcal{F}_t)_{t\in\mathbb{N}}$, for all $h \in \mathcal{H}$. Then, for any distribution $P_0$ on $\mathcal{H}$ and all $\delta \in (0, 1)$, we have*

$$\mathbf{P}\left[\forall n \geq 1, \forall P, \quad \frac{\mathbf{E}_{P(h)}[|S_n^h|]^2}{2(\rho+1)\mathbf{E}_{P(h)}[V_n^h] + 2\mathbf{E}_{P(h)}[|S_n^h|]} \leq \mathrm{KL}(P\|P_0) + \ln\frac{\phi_\rho(n)}{\delta}\right] \geq 1 - \delta, \quad (6)$$

*where $S_n^h := \sum_{t=1}^n X_t^h$ and $V_n^h := 1 + \sum_{t=1}^n (X_t^h)^2$.*

The proof of the theorem is in Appendix B. The bound in Proposition 5 does not look like the typical PAC-Bayesian bound. However, simple algebra reveals that for any $C > 0$,

$$\frac{\mathbf{E}_{P(h)}[|S_n^h|]^2}{2(\rho+1)\mathbf{E}_{P(h)}[V_n^h] + 2\mathbf{E}_{P(h)}[|S_n^h|]} \leq C \implies \mathbf{E}_{P(h)}[|S_n^h|] \leq 2C + \sqrt{2(\rho+1)\mathbf{E}_{P(h)}[V_n^h] \cdot C}.$$

Combining this with the fact that $|\mathbf{E}_{P(h)}[S_n^h]| \leq \mathbf{E}_{P(h)}[|S_n^h|]$ (by Jensen's inequality), and (6), we obtain, under the same conditions as Proposition 5 that

$$\mathbf{P}\left[\forall n \geq 1, \forall P \in \triangle(\mathcal{H}), \quad |\mathbf{E}_{P(h)}[S_n^h]| \leq 2C_n(P) + \sqrt{2(\rho+1)\mathbf{E}_{P(h)}[V_n^h] \cdot C_n(P)}\right] \geq 1 - \delta, \quad (7)$$

where $C_n(P) := \mathrm{KL}(P\|P_0) + \ln(\phi_\rho(n)/\delta)$. When $\mathcal{H}$ is a singleton, the concentration inequality in (7) can be viewed as an empirical version of Freedman's inequality [23] for MDS. The inequality is also reminiscent of the PAC-Bayesian empirical Bernstein inequality due to [50]. In addition to it holding for MDS, another advantage of (7) is that it is time-uniform—it holds for all sample sizes $n$, simultaneously. While standard concentration inequalities that hold for a fixed sample size can be turned into time-uniform ones by applying a union bound over sample sizes, the resulting inequalities will have sub-optimal $\ln n$ factors under the main square-root error term. In contrast, the concentration inequality in (7) has a $\ln(\phi_\rho(n)) = O(\ln\ln n)$ term, which matches the optimal dependence in $n$ according to the law of iterated logarithm [13]. MDS is a central concept in probability theory [44] and machine learning [10], and so our result in Proposition 5 is of independent interest.

Interestingly, the concentration inequality in Proposition 5 is derived using the guarantee of a parameter-free online algorithm—FREEGRAD. This opens the door for new ways of deriving such concentration inequalities through online learning algorithms, adding to existing results due to [45, 22, 28].

Using the result of Proposition 5, we are now going to derive a time-uniform "Empirical Bernstein" concentration inequality that holds under Assumption 1:

**Theorem 6.** *Let $\delta \in (0, 1)$, $\rho > 1$, and $\phi_\rho$ be as in (4). Further, for $P_0 \in \triangle(\mathcal{H})$ define $\epsilon_n(P) := \frac{2(\rho+1)}{n}(\mathrm{KL}(P\|P_0) + \ln\frac{\phi_\rho(n)}{\delta})$ and $\mathcal{P}_n := \{P : 1 - \epsilon_n(P) > 0\}$. Under Assumption 1, we have*

$$\mathbf{P}\left[\forall n \geq n_\delta, \forall P \in \mathcal{P}_n, \quad \left|\frac{1}{n}\sum_{t=1}^n \mathbf{E}_{P(h)}[\ell(h, Z_t) - L(h)]\right| \leq \frac{\sqrt{\epsilon_n(P) \cdot \widehat{V}_n(P)} + \frac{\epsilon_n(P)}{\rho+1}}{1 - \epsilon_n(P)}\right] \geq 1 - \delta,$$

*where $\widehat{V}_n(P) := \frac{1}{n}\sum_{t=1}^n \mathbf{E}_{P(h)}\left[\left(\ell(h, Z_t) - \frac{1}{n}\sum_{i=1}^n \mathbf{E}_{P(\theta)}[\ell(\theta, Z_i)]\right)^2\right] + \frac{1}{n}$ and $n_\delta$ is as in (5).*

The restriction that $P \in \mathcal{P}_n$ in the theorem merely ensures that the denominator in the concentration bound remains positive. The set $P_n$ is guaranteed to be non-empty for all $n \geq n_\delta$. In this case, $P$ is in

$\mathcal{P}_n$ whenever $8(\rho+1)\operatorname{KL}(P\|P_0) \le n$, which is a fairly weak condition on the distribution $P$; even for large models such as Neural Networks the KL-divergence $\operatorname{KL}(P\|P_0)$ does not typically grow superlinearly with the size $n$ of the sample used to generate the posterior $P$ [55]. Nevertheless, we note that if one only cares about the i.i.d. setting then other concentration inequalities may be used to achieve risk monotonicity without restrictions on the posterior $P$ (see Appendix D).

The concentration inequality in Theorem 6 can be viewed as an extension of the empirical Bernstein inequality in [35, 50] that holds under the non-i.i.d. condition described in Assumption 1. Our new bound is also time-uniform with the optimal dependence in the sample size $n$. As mentioned before, simply applying a union bound to a standard (non-time-uniform) concentration inequalities to obtain its time-uniform version will lead to a sub-optimal dependence in $n$. When $\mathcal{H}$ is a singleton (i.e. no learning), the concentration inequality becomes reminiscent of an existing one due to [26]. However, the latter has a term that looks like, but is different than, the empirical variance, and so it is not directly comparable to ours. The proof of Theorem 6 is postponed to Appendix B.

Finally, we note that for any sample size $n$ the value of $\rho$ in Theorem 6 that minimizes the bound would typically fall within the interval $(1,2)$. One can tune $\rho$ as a function of the data by treating $\rho$ as an extra "hypothesis" parameter. In the case of a finite grid $\mathcal{G} \subset (1,+\infty)$ of $\rho$'s, the result of Theorem 6 would hold for all $\rho \in \mathcal{G}$ inside the probability event as long as any $\operatorname{KL}(P\|P_0)$ instance is replaced by $\operatorname{KL}(P\|P_0) + \ln|\mathcal{G}|$.

We now move on to describing our risk monotonic procedure that makes use of our new concentration inequality.

## 4   Risk Monotonicity in Statistical Learning

In this section, we combine the concentration inequality from Theorem 6 with a novel "greedy" procedure for selecting distributions over hypotheses to derive a risk monotonic algorithm in the statistical learning setting. With the right choice of gap sequence $(\xi_n)$, the procedure we present in Algorithm 1 takes as input a base learning algorithm $\mathsf{B} : \bigcup_{i=1}^{\infty} \mathcal{Z}^i \to \triangle(\mathcal{H})$ together with samples $Z_{1:n}$ whose generating process satisfies Assumption 1, and returns a distribution over $\mathcal{H}$ that has a monotonic risk as a function of $n$ with high probability. By leveraging our new concentration inequality in Theorem 6 to specify the gap sequence $(\xi_n)$, we further show that achieving risk monotonicity need not deteriorate rates of convergence to the optimal risk $\inf_{h \in \mathcal{H}} L(h)$. In fact, we show that it is possible to attain fast rates under the Bernstein condition (Definition 3). To arrive at this result, it was crucial for our concentration inequality to have an empirical variance term.

---

**Algorithm 1** A Risk Monotonic Algorithm Wrapper

**Require:**
   A base learning algorithm $\mathsf{B} : \bigcup_{i=1}^{\infty} \mathcal{Z}^i \to \triangle(\mathcal{H})$.
   Samples $Z_1, \ldots, Z_n$.
   A sequence of gap functions $(\xi_k)$, where $\xi_k = \xi_k(Z_{1:k}, \mathsf{B})$ for all $k$.   ▷ $(\xi_k)$ will be set according to (9) [resp. (11)] for risk-monotonicity in probability [resp. expectation].

1: **for** $k = 1, \ldots, n$ **do**
2:   **if** $\dfrac{1}{k}\sum_{i=1}^{k} \mathbf{E}_{\mathsf{B}(h|Z_{1:k})}[\ell(h, Z_i)] - \dfrac{1}{k}\sum_{i=1}^{k} \mathbf{E}_{P_{k-1}(h)}[\ell(h, Z_i)] \le -\xi_k$ **then**
3:     Set $P_k = \mathsf{B}(Z_{1:k})$.
4:   **else**
5:     Set $P_k = P_{k-1}$.
6: Return $P_n$.

---

To simplify our analysis, we will focus our attention on base algorithms that are restricted in the following way:

**Assumption 2** (Base Algorithm Restriction). *We assume access to a base algorithm* $\mathsf{B} : \bigcup_{i=1}^{\infty} \mathcal{Z}^i \to \triangle(\mathcal{H})$ *such that for* $\rho > 1$*, prior distribution* $P_0$ *on* $\mathcal{H}$*, and all* $k \ge 1$*,*

$$\mathsf{B}(Z_{1:k}) \in \mathcal{Q}_k \coloneqq \{Q \in \triangle(\mathcal{H}) : 16(\rho+1)\operatorname{KL}(Q\|P_0) \le k\}. \tag{8}$$

In other words, we are restricting our attention to algorithms whose posteriors given samples of size $k$ do not have a KL divergence to the prior $P_0$ that grow too quickly with $k$. We make this assumption to satisfy the technical conditions needed for the concentration bound in Theorem 6 to be non-vacuous. Assumption 2 is reasonable even for large Neural Network models [55][4].

In practice, if $\mathcal{H}$ is $\mathbb{R}^d$, an option for B is the algorithm that outputs a multivariate Gaussian distribution around a regularized ERM. The KL-divergence between the outputs of B and $P_0$ can be controlled by tuning the regularisation parameter(s) and/or the co-variance matrix of the multivariate Gaussian.

**Remark 7.** *If one only cares about i.i.d. loss processes, Assumption 2 can be removed by using generalization bounds based on the standard Bernstein concentration inequality, e.g. [50, 36], which would also enable our risk decomposition in Theorem 9 below (and thus fast rates). We make Assumption 2 only to bound the denominator in our (non-i.i.d) concentration in Theorem 6 away from zero, which is not needed for other concentration bounds in the i.i.d. setting such as those in [50, 36] (see Appendix D for more detail).*

To specify the sequence of gaps $(\xi_n)$ that our wrapper Algorithm 1 requires, we will use our concentration bound in Theorem 6. We recall that using this concentration bound instead of other existing ones allows us to I) achieve risk monotonicity under a weaker condition than i.i.d. on the loss process (Assumption 1 in this case); and II) to achieve potentially fast excess risk rates under the Bernstein condition. The latter is made possible by the fact that the concentration bound in Theorem 6 has an empirical loss variance term that allows a particularly useful decomposition of the excess risk under the Bernstein condition (see Theorem 9 below).

To give a concise expression of the gaps $(\xi_k)$, we let $Q_k := \mathsf{B}(Z_{1:k}) \times P_{k-1}$, where $(P_k)$ are the intermediate distributions generated internally by Algorithm 1. With this, and the convention that $1/0 = +\infty$, we define

$$\xi_k := \frac{\sqrt{\epsilon_k \cdot \widehat{V}_k} + \frac{2\epsilon_k}{\rho+1}}{|1 - \epsilon_k|}, \quad \text{where} \quad \epsilon_k := \frac{2\,\mathrm{KL}(Q_k \| P_0 \times P_0) + 2\ln\frac{\phi_\rho(k)}{\delta}}{k \cdot (\rho+1)^{-1}}; \quad \text{and} \tag{9}$$

$$\widehat{V}_k := \frac{1}{k}\sum_{t=1}^{k} \mathbf{E}_{Q_k(h,h')}\left[\left(\ell(h, Z_t) - \ell(h', Z_t)\right)^2\right] - \left(\frac{1}{k}\sum_{t=1}^{k}\mathbf{E}_{Q_k(h,h')}\left[\ell(h, Z_t) - \ell(h', Z_t)\right]\right)^2.$$

We note that our Assumption 2 ensures that $\epsilon_n \leq 1/2$ (which in turn ensures that the gaps $(\xi_n)$ are not too large), for all $n \geq n_\delta$, where $n_\delta$ is as in (5). This follows from the fact that $\mathrm{KL}(Q_k \| P_0 \times P_0) = \mathrm{KL}(\mathsf{B}(Z_{1:k}) \| P_0) + \mathrm{KL}(P_{k-1} \| P_0)$, and that there exists $m < k$ such that $P_{k-1} = \mathsf{B}(Z_{1:m}) \in \mathcal{Q}_m \subseteq \mathcal{Q}_k$ (by definition of $(P_k)$ in Algorithm 1). Furthermore, $\mathcal{Q}_n \neq \varnothing$ for all $n \geq n_\delta$.

Before presenting our results for this section, we also note that Algorithm 1 assumes we can evaluate expectations over the output distributions of B. Such expectations can be approximated via Monte Carlo sampling, and any approximation errors need be added to the gaps $(\xi_n)$ to maintain the guarantees we present. Alternatively, one can avoid estimating expectations by applying recent derandomization techniques, see e.g. [46, 51], or by using other, non PAC-Bayesian generalization bounds (see Appendix D).

We now state the guarantees of Algorithm 1. We start by the statement of risk-monotonicity (the proof is postponed to Appendix C):

**Theorem 8** (Risk Monotonicity). *Let $\delta \in (0, 1)$ and $n_\delta$ be as in (5). Under Assumptions 1 and 2, Algorithm 1 with $(\xi_k)$ as in (9) is $(\delta, n_\delta)$-risk-monotonic according to Definition 1.*

We now show that risk monotonicity need not come at a worse excess risk rate. Under the Bernstein condition, we have the following excess risk decomposition for the output of Algorithm 1:

**Theorem 9** (Risk Decomposition). *Let $B > 1$, $\beta \in [0, 1]$, and suppose that the $(\beta, B)$-Bernstein condition holds. Further, for $\rho > 1$ and $\delta \in (0, 1)$, let $n_\delta$ and $\epsilon_n$ be as in (5) and (9), respectively. Then, under Assumptions 1 and 2, the outputs $(P_k)$ of Alg. 1 satisfy, with probability at least $1 - 2\delta$,*

$$\forall n \geq n_\delta, \quad L(P_n) - L(h_\star) \leq 3\big(L(\mathsf{B}(Z_{1:n})) - L(h_\star)\big) + O\left(\epsilon_n\right)^{\frac{1}{2-\beta}}, \tag{10}$$

*where $h_\star \in \arg\inf_{h \in \mathcal{H}} L(h)$.*

---

[4]Besides, when the KL divergence is very large, it is often very hard to infer non-vacuous generalization bounds (which one may view as a prerequisite to achieving risk monotonicity) using a PAC-Bayesian approach.

We stress that this risk decomposition was only made possible by the fact that our concentration bound in Theorem 6 has an empirical loss variance term.

Theorem 9 shows that the excess risk of Algorithm 1 is at most a constant times the excess risk of the base algorithm B, plus a potentially lower-order term $O(\epsilon_n^{\frac{1}{2-\beta}})$. To appreciate what this additional term is doing, consider the case of a finite hypothesis class $\mathcal{H}$. In this case, the definition of $\epsilon_n$ in (9) implies that $\epsilon_n \leq O(\frac{1}{n} \ln \frac{|\mathcal{H}| \ln n}{\delta})$. Thus, $\epsilon_n^{\frac{1}{2-\beta}}$ interpolates between the fast $\frac{1}{n} \ln \frac{|\mathcal{H}| \ln n}{\delta}$ rate under the best Bernstein condition with $\beta = 1$ and the standard (up to log-log-factors) rate $\sqrt{\frac{1}{n} \ln \frac{|\mathcal{H}| \ln n}{\delta}}$ under the Bernstein condition with $\beta = 0$, which we recall always holds for bounded losses. What is more, if $\mathcal{H}$ is finite and algorithm B is the ERM, i.e. if $B(Z_{1:k})$ is a Dirac at $\hat{h}_n \in \arg\inf_{h \in \mathcal{H}} \sum_{i=1}^{n} \ell(h, Z_i)$, then we have the following explicit excess risk rate for Algorithm 1:

**Proposition 10.** *Under the setting of Theorem 9, if $\mathcal{H}$ is finite, $P_0$ is set to the uniform prior over $\mathcal{H}$, and $B(Z_{1:n})$ is a point mass around the ERM $\hat{h}_n \in \arg\min_{h \in \mathcal{H}} \sum_{i=1}^{n} \ell(h, Z_i)$, $\forall n \geq 1$, then Alg. 1 is risk monotonic according to Def. 1 and its outputs $(P_k)$ satisfy, with probability at least $1 - 2\delta$,*

$$\forall n \geq n_\delta \vee (16(\rho+1)\ln|\mathcal{H}|), \quad L(P_n) - L(h_\star) \leq O\left(\frac{\ln \frac{|\mathcal{H}|\ln n}{\delta}}{n}\right)^{\frac{1}{2-\beta}}.$$

The story is not much different for a continuous set $\mathcal{H}$. The standard excess risk rate one would expect from algorithm B is $\sqrt{(\mathrm{KL}(B(Z_{1:k})\|P_0) + \ln(1/\delta))/k}$, which can dominate the right-most term in our risk decomposition (10) since[5]

$$(\epsilon_n)^{\frac{1}{2-\beta}} \leq O\left(\frac{\max_{k \leq n} \mathrm{KL}(B(Z_{1:k})\|P_0) + \ln \frac{\ln n}{\delta}}{n}\right)^{\frac{1}{2-\beta}}.$$

Together, the above inequality and Theorem 9 show that fast rates for algorithm 1 are achievable whenever the Bernstein condition holds with $\beta > 0$ and the base algorithm B itself achieves a fast rate (when $\mathcal{H}$ is finite and B is the ERM, Proposition 10 shows that it is sufficient that $\beta > 0$). Thus, risk monotonicity need not come at the price of a worse rate of convergence to the optimal risk. Finally, we note that the factor 3 in our risk decomposition (10) is just an artifact of our analysis. In fact, by slightly modifying our proof of Theorem 9, one can show that any factor in the interval $(1, 3]$ is achievable at the cost of a larger lower-order term in (10).

**Risk monotonicity in expectation.** The original open problem due to [53] was around risk monotonicity in expectation—as in (3). Our algorithm wrapper also allows us to almost[6] achieve this notion of monotonicity with a slight modification of the sequence $(\xi_k)$ in (9); we essentially set the confidence level $\delta$ as a function of $k$. In fact, the next result shows that risk monotonicity in expectation is achievable up to an additive fast-rate term:

**Theorem 11.** *Let $b \geq 1$, $\rho > 1$, and $\phi_\rho$ be as in (4). Under Assumptions 1 and 2, Algorithm 1 with the sequence of gaps $(\xi_k)$ given by*

$$\xi_k = \xi_k' := \frac{1}{|1 - \epsilon_k'|}\left(\sqrt{\epsilon_k' \cdot \widehat{V}_k} + \frac{2\epsilon_k'}{\rho+1}\right), \tag{11}$$

*with $\epsilon_k' := \frac{2(\rho+1)}{k}\left(\mathrm{KL}(B(Z_{1:k}) \times P_{k-1}\|P_0 \times P_0) + 2\ln(\phi_\rho(k)) + 2b\ln k\right)$ and $(P_k)$ being as in Algorithm 1, satisfies, for all $t \geq N := \sup\{n : 8\ln(n^b\phi_\rho(n)) > n\}$,*

$$\mathbf{E}\left[\mathbf{E}_{P_t(h)}[L(h)]\right] \leq \mathbf{E}\left[\mathbf{E}_{P_{t-1}(h)}[L(h)]\right] + 1/t^b. \tag{12}$$

We note that $N$ need not be too large since $\phi_\rho(n) \leq O(\ln n)$. However, while taking $b$ large will make the additive $1/t^b$ term in (12) small, it will increase the sample size $N$ beyond which (12) holds.

---

[5] We note that it is typical that $\max_{k \leq n} \mathrm{KL}(B(Z_{1:k})\|P_0) \leq O(\mathrm{KL}(B(Z_{1:n})\|P_0))$, for $n \geq 1$.

[6] We thank Olivier Bousquet and his colleagues for pointing out a mistake in the original proof of Theorem 11. The theorem originally claimed that Algorithm 1 achieves risk-monotonicity in expection without the additive $1/t^b$ term in (12), which turns out not to be correct.

Ignoring the additive $1/t^b$ term, the notion of risk-monotonicity in theorem 11 corresponds to the notion of weak $\mathcal{Z}$-monotonicity in [33]—one of their strongest notions of risk monotonicity since it holds for all data generating distributions (granted Assumption 1 holds, or the data is i.i.d.). We can modify Algorithm 1 to make it globally $\mathcal{Z}$-monotonic, where (12) would hold for all $t \geq 1$ (instead of $t \geq N$), by forcing the outputs $P_k$ of the algorithm to be $P_0$ for all $k < N$.

With the same choice of gap sequence $(\xi_k)$ in Theorem 11 it is easy to show (following almost the same proof steps as in our previous results) that the excess risk rates in Theorem 9 and Proposition 10 will hold with probability $1/n^b$ (instead of $\delta$) for every sample size $n$.

## 5 Discussion and Future Work

The primary goal of this paper was to answer the fundamental question around the existence of a consistent, risk-monotonic algorithm in the general statistical learning setting. We answer this in the affirmative for a notation of monotonicity that holds with high probability and further show that there is virtually no cost for achieving this when it comes to excess-risk rates. We believe this is an important milestone in the search for risk-monotonic algorithms. It remains to see if risk-monotonicity in expectation is achievable; i.e. achieving (12) without the additive term.

From a computational perspective, the main setback of Algorithm 1 is that returning the distribution $P_n = P_n(Z_{1:n})$ requires $n$-calls to the base algorithm B. This implies that 1 may run $n$ times slower than B in the worst case. However, when the base algorithm outputs distributions centered at ERMs, it may be possible to efficiently generate the sequence of distributions $(B(Z_{1:k}))_k$ by leveraging the fact that ERM solutions for sample sizes $k$ and $k + 1$ can be close to each other.

When the loss $\ell$ is convex in the first argument, there is no need for a randomized algorithm[7] (i.e. we do not need $B(Z_{1:k}) \ll P_0$), and it is possible to efficiently generate a sequence of predictors $(\hat{h}_n)$ with monotonic risk using an online convex optimization algorithm as the base algorithm B. However, in general, it is unclear whether risk-monotonicity can be achieved without the (greedy) for-loop procedure of Algorithm 1. We note also that if one only wants a decreasing risk after some sample size $s \in \mathbb{N}$, then computing the distributions $(P_k)_{k<s}$ is unnecessary. In this case, the for-loop in Algorithm 1 need only start at $k = s$; the resulting hypotheses would satisfy the monotonicity condition in Definition 1 for all $n \geq s \vee n_\delta$.

Through Theorem 9, we showed that the excess risk of Algorithm 1 is at most three times that of the base algorithm plus a lower-order term. It remains to evaluate the empirical performance of the algorithm to identify for which applications the additional cost in the excess risk is worth it to achieve risk monotonicity.

Some important questions remain open along the axes of assumptions. In particular, can we remove the boundedness condition on the loss while retaining risk-monotonicity? It might be possible to achieve this for unbounded convex losses using the non-exponential weighted aggregation techniques recently suggested by [2]. Lifting the boundedness assumption may be key in resolving another COLT open problem [24] regarding achievable risk rates of log-loss Bayesian predictors. Our results build foundations for these avenues, which are promising subjects for future work.

Finally, it would be interesting to explore what other concentration inequalities for non-i.i.d. processes can be derived using other parameter-free online learning algorithms. An obvious starting point is to look at MATRIX-FREEGRAD [39].

---

[7]Randomization is also not needed if one is only interested in i.i.d. processes (see Appendix D).

## Acknowledgments and Disclosure of Funding

We would like to thank Hisham Husain for instrumental discussions during the early phases of the project. We thank Olivier Bousquet and his colleagues for pointing out a mistake in the original proof of Theorem 11. We also thank anonymous reviewers for their valuable feedback. This work was supported by the Australian Research Council and Data61.

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
