## A  Technical Results

In this section, it will be convenient to adopt the ESI notation [29]:

**Definition 12** (Exponential Stochastic Inequality (ESI) notation). *Let $(\Omega, \mathcal{F}, \mathbf{P})$ be a probability space. Further, let $X$, $Y$ be any two random variables and $\mathcal{G}$ be a sub-$\sigma$-algebra of $\mathcal{F}$. For $\eta > 0$, we define*

$$X \trianglelefteq_\eta^{\mathcal{G}} Y \iff X - Y \trianglelefteq_\eta^{\mathcal{G}} 0 \iff \mathbf{E}\left[e^{\eta(X-Y)} \mid \mathcal{G}\right] \leq 1.$$

For $\mathcal{G} = \mathcal{F}$, we simply write $\trianglelefteq_\eta$ instead of $\trianglelefteq_\eta^{\mathcal{G}}$. In what follows, given random variables $Z_1, Z_2, \ldots$ and loss $\ell$ satisfying Assumption 1, we denote by

$$X_i^h \coloneqq \ell(h, Z_i) - \ell(h_*, Z_i), \quad h \in \mathcal{H}, i \in \mathbb{N},$$

the excess-loss random variable, where $h_* \in \arg\inf_{h \in \mathcal{H}} L(h)$ (with $L$ as in Assumption 1). Let

$$\Phi_{i,\eta} \coloneqq \frac{1}{\eta} \ln \mathbf{E}_{i-1}\left[e^{-\eta X_i^h}\right] = \frac{1}{\eta} \ln \mathbf{E}\left[e^{-\eta X_i^h} \mid Z_1, \ldots, Z_{i-1}\right] \tag{13}$$

be the (conditional) *normalized cumulant generating function* of $X_i^h$. We note that since the loss $\ell$ takes values in the interval $[0, 1]$, we have

$$X_i^h \in [-1, 1], \quad \text{for all } h \in \mathcal{H}, \text{a.s.}$$

We now present some existing results pertaining to the excess-loss random variable $X_i^h$ and its normalized cumulant generating function, which will be useful in our proofs:

**Lemma 13** ([29]). *Let $h \in \mathcal{H}$ and $i \in \mathbb{N}$. Further, let $X_i^h$, and $\Phi_{i,\eta}$ be as above. Then, for all $\eta \geq 0$,*

$$\alpha_\eta \cdot (X_i^h)^2 - X_i^h(Z) \trianglelefteq_\eta^{\mathcal{G}_{i-1}} \Phi_{i,2\eta} + \alpha_\eta \cdot \Phi_{i,2\eta}^2, \quad \text{where } \alpha_\eta \coloneqq \frac{\eta}{1 + \sqrt{1 + 4\eta^2}},$$

*and $\mathcal{G}_{i-1}$ is the $\sigma$-algebra generated by $Z_1, \ldots, Z_{i-1}$.*

**Lemma 14** ([29]). *If the $(\beta, B)$-Bernstein condition (Definition 3) holds for $(\beta, B) \in [0, 1] \times \mathbb{R}_{>0}$, then for $\Phi_{i,\eta}$ as in (13), it holds that*

$$\Phi_{i,\eta} \leq (B\eta)^{\frac{1}{1-\beta}}, \quad \text{for all } \eta \in (0, 1], i \geq 1.$$

**Lemma 15** ([10]). *For $\Phi_{i,\eta}$ as in (13), it holds that*

$$\Phi_{i,\eta} \leq \frac{\eta}{2}, \quad \text{for all } \eta \in \mathbb{R}, i \geq 1.$$

**Lemma 16** ([10]). *For $i \geq 1$ and $h \in \mathcal{H}$, the excess-loss random variable $X_i^h$ satisfies*

$$X_i^h - \mathbf{E}_{i-1}[X_i^h] \trianglelefteq_\eta^{\mathcal{G}_{i-1}} \eta \cdot \mathbf{E}_{i-1}[(X_i^h)^2], \quad \text{for all } \eta \in [0, 1],$$

*where $\mathcal{G}_{i-1}$ is the $\sigma$-algebra generated by $Z_1, \ldots, Z_{i-1}$ and $\mathbf{E}_{i-1}[\cdot] \coloneqq \mathbf{E}[\cdot \mid \mathcal{G}_{i-1}]$.*

The following useful proposition is imported from [38] with minor modifications:

**Proposition 17. [ESI Transitivity]** *Let $(\Omega, \mathcal{F}, \mathbf{P})$ be a probability space and $\mathcal{G}$ be a sub-$\sigma$-algebra of $\mathcal{F}$. Further, let $Z_1, \ldots, Z_n$ be random variables such that for $(\gamma_i)_{i \in [n]} \in (0, +\infty)^n$, $Z_i \trianglelefteq_{\gamma_i}^{\mathcal{G}} 0$, for all $i \in [n]$. Then*

$$\sum_{i=1}^n Z_i \trianglelefteq_{\nu_n}^{\mathcal{G}} 0, \quad \text{where } \nu_n \coloneqq \left(\sum_{i=1}^n \frac{1}{\gamma_i}\right)^{-1}.$$

To prove our time-uniform concentration inequality in Section 3, we will require the following generalization of Markov's inequality (we state the version found in [27]):

**Lemma 18** (Ville's inequality). *If $(M_n)_{n \geq 0}$ is a non-negative supermartingale, then for any $a > 0$,*

$$\mathbf{P}[\exists n \geq 1 : M_n \geq a] \leq \frac{M_0}{a}.$$

The upcoming lemmas will help us bound the sequence of gaps $(\xi_k)$ in (9) under the Bernstein condition.

**Lemma 19.** *Let $P_0 \in \triangle(\mathcal{H})$, $\beta \in [0,1]$ and $B > 0$, and suppose that the $(\beta, B)$-Bernstein condition holds. Then, under Assumption 1, for any $\eta \in [0, 1/2]$ and $\delta \in (0,1)$, with probability at least $1 - \delta$,*

$$\frac{\eta}{n} \sum_{i=1}^{n} \mathbf{E}_{Q(h)}[(\ell(h, Z_i) - \ell(h_\star, Z_i))^2] \leq 8(L(Q) - L(h_\star)) + 4C_\beta \cdot \eta^{\frac{1}{1-\beta}}$$
$$+ \frac{8(\mathrm{KL}(Q\|P_0) + \ln \delta^{-1})}{n\eta}, \tag{14}$$

*for all $n \geq 1$, where $h_\star \in \arg\inf_{h \in \mathcal{H}} L(h)$ and $C_\beta := \left((1-\beta)^{1-\beta}\beta^\beta\right)^{\frac{\beta}{1-\beta}} + 3/2(2B)^{\frac{1}{1-\beta}}$.*

**Proof of Lemma 19.** Let $\delta \in (0,1)$ and define $X_i^h := \ell(h, Z_i) - \ell(h_\star, Z_i)$. We recall that $\mathcal{G}_i$ is the $\sigma$-algebra generated by $Z_1, \ldots, Z_i$, and $\mathbf{E}_{i-1}[\cdot] := \mathbf{E}[\cdot \mid \mathcal{G}_{i-1}]$. Note that under Assumption 1, $\mathbf{E}_{i-1}[X_i^h] = L(h) - L(h_\star)$, for all $i \geq 1$ and $h \in \mathcal{H}$. For any $\eta \in [0, 1/2]$ and $h \in \mathcal{H}$ our strategy is to show that, under the $(\beta, B)$-Bernstein condition,

$$M_n^h := \exp\left(\eta^2 \sum_{i=1}^{n} (X_i^h)^2/8 - n\eta \cdot (L(h) - L(h_\star)) + nC_\beta \cdot \eta^{\frac{2-\beta}{1-\beta}}/2\right), \tag{15}$$

is a non-negative supermartingale, for all $h \in \mathcal{H}$. After that, invoking Ville's inequality (Lemma 18) and applying a change of measure argument (Lemma 21), we get the desired result.

Under the $(\beta, B)$-Bernstein condition, Lemmas 13-15 imply, for all $\eta \in [0, 1/2]$ and $i \geq 1$,

$$\eta \cdot (X_i^h)^2/4 \trianglelefteq_\eta^{\mathcal{G}_{i-1}} X_i^h + 3/2 (2B\eta)^{\frac{1}{1-\beta}}, \tag{16}$$

where we used the fact that $\alpha_\eta = \frac{\eta}{1+\sqrt{1+4\eta^2}} \geq \eta/4$, for all $0 \leq \eta \leq 1/2$ ($\alpha_\eta$ is involved in Lemma 13). Now, due to the Bernstein inequality (Lemma 16), we have for all $\eta \in [0, 1/2]$ and $i \geq 1$,

$$X_i^h \trianglelefteq_\eta^{\mathcal{G}_{i-1}} L(h) - L(h_\star) + \eta \cdot \mathbf{E}_{i-1}[(X_i^h)^2],$$
$$\trianglelefteq_\eta^{\mathcal{G}_{i-1}} L(h) - L(h_\star) + \eta \cdot (L(h) - L(h_\star))^\beta, \quad \text{(by the Bern. cond. \& Assumption 1)}$$
$$\trianglelefteq_\eta^{\mathcal{G}_{i-1}} 2(L(h) - L(h_\star)) + c_\beta^{\frac{\beta}{1-\beta}} \cdot \eta^{\frac{1}{1-\beta}}, \quad \text{where } c_\beta := (1-\beta)^{1-\beta}\beta^\beta. \tag{17}$$

The last inequality follows by the fact that $z^\beta = c_\beta \cdot \inf_{\nu > 0}\{z/\nu + \nu^{\frac{\beta}{1-\beta}}\}$, for $z \geq 0$ (in our case, we set $\nu = c_\beta\eta$ to get to (17)). By chaining (16) with (17) using Proposition 17, we get:

$$\eta \cdot (X_i^h)^2/4 \trianglelefteq_{\eta/2}^{\mathcal{G}_{i-1}} 2(L(h) - L(h_\star)) + c_\beta^{\frac{\beta}{1-\beta}} \cdot \eta^{\frac{1}{1-\beta}} + 3/2(2B\eta)^{\frac{1}{1-\beta}}.$$
$$\trianglelefteq_{\eta/2}^{\mathcal{G}_{i-1}} 2(L(h) - L(h_\star)) + C_\beta \cdot \eta^{\frac{1}{1-\beta}}. \tag{18}$$

This implies that $M_n^h$ in (15) is a non-negative supermartingale. This in turn implies that for any distribution $P_0$, $\mathbf{E}_{P_0(h)}[M_n^h]$ is also a supermartingale. Thus, by Ville's inequality (Lemma 18), we have, for any $\delta \in (0,1)$,

$$\delta \geq \mathbf{P}\left[\exists n \geq 1, \mathbf{E}_{P_0(h)}[M_n^h] \geq \delta^{-1}\right], \tag{19}$$

On the other hand, by the KL-change of measure lemma (Lemma 21), we have for all $Q \in \triangle(\mathcal{H})$

$$\mathbf{E}_{Q(h)}[\ln M_n^h] \leq \mathrm{KL}(Q\|P_0) + \mathbf{E}_{P_0(h)}[M_n^h].$$

Combining this with (19), we get the desired result. $\qquad \square$

**Lemma 20.** *For $A, B > 0$, we have*

$$\inf_{\eta \in (0, 1/2)}\left\{A\eta^{\frac{1}{1-\beta}} + B/\eta\right\} \leq \frac{A(3-2\beta)}{1-\beta}\left(\frac{(1-\beta)B}{A}\right)^{\frac{1}{2-\beta}} + 2B. \tag{20}$$

*Proof.* The unconstrained minimizer of the LHS of (20) is given by $\eta_\star \coloneqq \left(\frac{(1-\beta)B}{A}\right)^{\frac{1-\beta}{2-\beta}}$. If $\eta_\star \leq 1/2$, then

$$\inf_{\eta \in (0,1/2]}\left\{A\eta^{\frac{1}{1-\beta}} + B/\eta\right\} \leq A\eta_\star^{\frac{1}{1-\beta}} + B/\eta_\star = \frac{A(2-\beta)}{1-\beta}\left(\frac{(1-\beta)B}{A}\right)^{\frac{1}{2-\beta}}. \tag{21}$$

Now if $\eta_\star > 1/2$, we have $(1/2)^{\frac{1}{1-\beta}} < \left(\frac{(1-\beta)B}{A}\right)^{\frac{1}{2-\beta}}$, and so, we have

$$\inf_{\eta \in (0,1/2]}\left\{A\eta^{\frac{1}{1-\beta}} + B/\eta\right\} \leq A(1/2)^{\frac{1}{1-\beta}} + 2B,$$

$$\leq A\left(\frac{(1-\beta)B}{A}\right)^{\frac{1}{2-\beta}} + 2B. \tag{22}$$

By combining (21) and (22) we get the desired result. □

We need one more classical change of measure result (see e.g. [1]):

**Lemma 21** (KL-**change of measure**). *For all distributions $P$ and $Q$ such that $Q \ll P$, it holds that*

$$\mathbf{E}_Q[X] \leq \inf_{\eta > 0}\left\{\eta\mathrm{KL}(Q\|P) + \eta^{-1}\ln\mathbf{E}_P\left[e^{\eta \cdot X}\right]\right\}.$$

# B  Proofs of the New Concentration Inequalities

To prove our first concentration inequality for MDS in Proposition 5, we start by constructing a non-negative supermartingale with the help of the recent FREEGRAD algorithm [39]. As mentioned in the introduction, our proof technique is similar to the one introduced in [28] with the difference that we use the specific shape of FREEGRAD's potential function to build our supermartingale. Using the latter leads to a desirable empirical variance term in our final concentration bound.

To express the FREEGRAD supermartingale, we define

$$\Phi_\gamma(S, Q) \coloneqq \frac{\gamma}{\sqrt{\gamma^2 + Q}} \cdot \exp\left(\frac{|S|^2}{2\gamma^2 + 2Q + 2|S|}\right), \quad S, Q \geq 0, \gamma > 0. \tag{23}$$

**Proposition 22.** *Let $\gamma > 0$ and $(\mathcal{F}_t)_{t \in \mathbb{N}}$ be a filtration. For any random variables $X_1, X_2, \cdots \in [-1, 1]$ s.t. $X_i$ is $\mathcal{F}_i$-measurable and $\mathbf{E}[X_i \mid \mathcal{F}_{i-1}] = 0$, for all $i \in [n]$, the process $(\Phi_\gamma(S_n, Q_n))$, where $S_n \coloneqq \sum_{i=1}^n X_i$ and $Q_n \coloneqq \sum_{i=1}^n X_i^2$ is a non-negative supermartingale w.r.t. $(\mathcal{F}_t)_{t \in \mathbb{N}}$; that is,*

$$\Phi_\gamma(S_n, Q_n) \geq 0, \quad \text{and} \quad \mathbf{E}[\Phi_\gamma(S_{n+1}, Q_{n+1}) \mid \mathcal{F}_n] \leq \Phi_\gamma(S_n, Q_n), \quad \text{for all } n \geq 1.$$

As mentioned above, the proof of the proposition is based on the guarantee of the parameter-free online learning algorithm FREEGRAD. The algorithm operates in rounds, where at each round $t$, FREEGRAD outputs $\widehat{w}_t$ (that is a deterministic function of the past) in some convex set $\mathcal{W}$, say $\mathbb{R}^d$, then observes a vector $g_t \in \mathbb{R}^d$, typically the sub-gradient of a loss function at round $t$. The algorithm guarantees a regret bound of the form $\sum_{t=1}^T g_t^\top(\widehat{w}_t - w) \leq \widetilde{O}(\|w\|\sqrt{Q_T})$, for all $w \in \mathcal{W}$, where $Q_T \coloneqq \sum_{t=1}^T \|g_t\|^2$. What is more, FREEGRAD's outputs $(\widehat{w}_t)$ ensure the following (see [39, Theorem 5]):

$$\widehat{w}_t^\top g_t + \Phi_\gamma(S_t, Q_t) \leq \Phi_\gamma(S_{t-1}, Q_{t-1}), \tag{24}$$

where $S_t \coloneqq \|\sum_{i=1}^t g_i\|$ and $Q_t \coloneqq \sum_{i=1}^t \|g_i\|^2$. In the proof of Proposition 22, we will reason about the outputs of FREEGRAD in one dimension (i.e. $d = 1$) in response to the inputs $(g_t) \equiv (X_t)$.

One way to prove Proposition 22 is to show that FREEGRAD is a betting algorithm that bets fractions smaller than one of its current wealth at each round. In this case, Proposition 22 would follow from existing results due to, for example, [28]. However, for the sake of simplicity, we decided to present a proof that does not explicitly refer to bets.

**Proof of Proposition 22.** By [39, Theorem 5 and proof of Theorem 20], FREEGRAD's outputs $(\widehat{w}_i)$ in response to $(X_i)$ and parameter $\gamma > 0$ (playing the role of $1/\epsilon$ in their Theorem 20) guarantee[8],

$$\widehat{w}_{n+1} \cdot X_{n+1} + \Phi_\gamma(S_{n+1}, Q_{n+1}) \le \Phi_\gamma(S_n, Q_n), \quad \text{for all } n \in \mathbb{N},$$

Re-arranging this inequality and taking the expectation $\mathbf{E}[\cdot \mid \mathcal{F}_n]$ yields

$$\mathbf{E}[\Phi_\gamma(S_{n+1}, Q_{n+1}) - \Phi_\gamma(S_n, Q_n) \mid \mathcal{F}_n] \le -\mathbf{E}[\widehat{w}_{n+1} \cdot X_{n+1} \mid \mathcal{F}_n] = -\widehat{w}_{n+1} \cdot \mathbf{E}[X_{n+1} \mid \mathcal{F}_n] = 0,$$

where the penultimate equality follows by the fact that $\widehat{w}_{n+1}$ is a deterministic function of the history up to round $n$, and so it is $\mathcal{F}_n$-measurable. Finally, the last equality follows by the assumption that $\mathbf{E}[X_{n+1} \mid \mathcal{F}_n] = 0$. $\square$

Next, using standard tools from PAC-Bayesian analyses, we extend the result of Proposition 22 by allowing the random variables $(X_t)$ to depend on $h \in \mathcal{H}$. We will also "mix" over the free parameter $\gamma$ to obtain the optimal (doubly-logarithmic) dependence in $n$ in our final concentration bounds.

**Proposition 23.** *Let $(\mathcal{F}_t)_{t \in \mathbb{N}}$ be a filtration and $\{X_t^h\}$ be a family of random variables in $[-1, 1]$ s.t. $X_t^h$ is $\mathcal{F}_t$-measurable and $\mathbf{E}[X_t^h \mid \mathcal{F}_{t-1}] = 0$, for all $t \ge 1$ and $h \in \mathcal{H}$. Further, let $\pi$ and $P_0$ be prior distributions on $\mathbb{R}_{>0}$ and $\mathcal{H}$, respectively. Then, for any $\delta \in (0, 1)$, we have*

$$\mathbf{P}\left[\forall n \ge 1, \forall P \in \triangle(\mathcal{H}), \ \mathbf{E}_{P(h)}\left[\ln \mathbf{E}_{\pi(\gamma)}\left[\Phi_\gamma(S_n^h, Q_n^h)\right]\right] \le \mathrm{KL}(P \| P_0) + \ln(1/\delta)\right] \ge 1 - \delta,$$

*where $S_n^h := \sum_{i=1}^n X_i^h$ and $Q_n^h := \sum_{i=1}^n (X_i^h)^2$.*

**Proof of Proposition 23.** By the KL-change of measure lemma (Lemma 21), we have

$$\mathbf{E}_{P(h)}\left[\ln \mathbf{E}_{\pi(\gamma)}\left[\Phi_\gamma(S_n^h, Q_n^h)\right]\right] \le \mathrm{KL}(P \| P_0) + \ln \mathbf{E}_{P_0(h)}\mathbf{E}_{\pi(\gamma)}\left[\Phi_\gamma(S_n^h, Q_n^h)\right], \quad (25)$$

for all $n \ge 1$ and $P \in \triangle(\mathcal{H})$. On the other hand, by Proposition 22, we know that the process $(\Phi_\gamma(S_n^h, Q_n^h))$ is a supermartingale for any $\gamma > 0$. This in turn implies that $(\mathbf{E}_{P_0(h)}\mathbf{E}_{\pi(\gamma)}\left[\Phi_\gamma(S_n^h, Q_n^h)\right])_n$ is also a non-negative supermartingale, since a mixture of supermartingales is also a supermartingale. Now, by Ville's inequality (Lemma 18), we have, for all $\delta \in (0, 1)$,

$$\mathbf{P}\left[\forall n \ge 1, \ \mathbf{E}_{P_0(h)}\mathbf{E}_{\pi(\gamma)}\left[\Phi_\gamma(S_n^h, Q_n^h)\right] \le 1/\delta\right] \ge 1 - \delta.$$

By combining this inequality with (25), we obtain the desired result. $\square$

We now use Proposition 23 to prove Proposition 5 (some of the steps in the next proof are similar to ones found in [28]):

**Proof of Proposition 5.** Let $\rho > 1$ and $Q_n^h := \sum_{i=1}^n (X_i^h)^2$. We will apply Proposition 23 with a specific choice of prior $\pi$. In particular, we let $\pi$ be a prior on $\{\rho^{k/2} : k \ge 1\}$, such that for $k \ge 1$,

$$\pi(\rho^{k/2}) := \pi_k := \frac{1}{ck \ln^2(k+1)},$$

where $c$ is as in (4). For $n \ge 1$ and $h \in \mathcal{H}$, let $k_n \ge 1$ be such that

$$\rho^{k_n - 1} \le 1 \vee Q_n^h \le \rho^{k_n}. \quad (26)$$

Note that $k_n$ is guaranteed to exist and (26) implies that $k_n \le \ln_\rho(1 \vee Q_n^h) + 1 \le \ln_\rho(n) + 1$. Let $\gamma_n := \rho^{k_n/2}$. With our choice of $\pi$, we have, for all $h \in \mathcal{H}$,

$$\ln \mathbf{E}_{\pi(\gamma)}\left[\Phi_\gamma(S_n^h, Q_n^h)\right] \ge \ln \Phi_{\gamma_n}(S_n^h, Q_n^h) + \ln \pi(\gamma_n),$$

$$\ge \frac{|S_n^h|^2}{2\gamma_n + 2Q_n^h + 2|S_n^h|} + \ln\left(\frac{\gamma_n}{\sqrt{\gamma_n^2 + Q_n^h}}\right) + \ln \pi(\gamma_n),$$

$$\ge \frac{|S_n^h|^2}{2(\rho+1)(1 \vee Q_n^h) + 2|S_n^h|} - \ln\sqrt{\rho+1} + \ln \pi(\gamma_n), \quad (27)$$

$$\ge \frac{|S_n^h|^2}{2(\rho+1)V_n^h + 2|S_n^h|} - \ln\left(c\sqrt{\rho+1}(\ln_\rho(n) + 1)\ln^2(\ln_\rho(n) + 2)\right), \quad (28)$$

$$= 4\sup_{\eta \ge 0}\left\{\eta|S_n^h| - 2\eta^2(\rho+1)V_n^h - 2\eta^2|S_n^h|\right\} - \ln \phi_\rho(n), \quad (29)$$

---

[8]Technically, FREEGRAD also requires a sequence of hints $(h_t)$ that provides upper bounds on $(|X_t|)$. Since $X_i \in [-1, 1]$, these hints can all be set to 1.

where in (27) we used (26) and in (28) we used the fact that $k_n \leq 1 + \ln_\rho(n)$. Now, by an application of Jensen's inequality, we get from (29) that

$$\mathbf{E}_{P(h)}\left[\ln \mathbf{E}_{\pi(\gamma)}\left[\Phi_\gamma(S_n^h, Q_n^h)\right]\right] \geq 4 \sup_{\eta \geq 0}\left\{\eta \mathbf{E}_{P(h)}|S_n^h| - 2\eta^2(\rho+1)\mathbf{E}_{P(h)}[V_n^h] - 2\eta^2 \mathbf{E}_{P(h)}|S_n^h|\right\}$$
$$- \ln \phi_\rho(n),$$
$$= \frac{(\mathbf{E}_{P(h)}|S_n^h|)^2}{2(\rho+1)\mathbf{E}_{P(h)}[V_n^h] + 2\mathbf{E}_{P(h)}|S_n^h|} - \ln \phi_\rho(n).$$

Thus, we have $\mathbf{E}_{P(h)}\left[\ln \mathbf{E}_{\pi(\gamma)}\left[\Phi_\gamma(S_n^h, Q_n^h)\right]\right] \leq \mathrm{KL}(P\|P_0) + \ln(1/\delta)$ only if

$$\frac{(\mathbf{E}_{P(h)}[|S_n^h|])^2}{2(\rho+1)\mathbf{E}_{P(h)}[V_n^h] + 2\mathbf{E}_{P(h)}[|S_n^h|]} \leq \mathrm{C}_n(P) \coloneqq \mathrm{KL}(P\|P_0) + \ln \frac{\phi_\rho(n)}{\delta}.$$

Combining this fact with Proposition 23 implies the desired result. $\qquad \square$

**Proof of Theorem 6.** We will apply Proposition 5 with $X_t^h \coloneqq \ell(h, Z_t) - \mathbf{E}_{t-1}[\ell(h, Z_t)] = \ell(h, Z_t) - L(h)$, where the last equality follows by Assumption 1. As before, we let $S_n^h \coloneqq \sum_{i=1}^n X_i^h$ and $V_n^h \coloneqq 1 + \sum_{i=1}^n (X_i^h)^2$. By the classical bias-variance decomposition, we have

$$\mathbf{E}_{P(h)}[V_n^h] = n\widehat{V}_n(P) + \mathbf{E}[S_i^h]^2/n, \tag{30}$$

where $\widehat{V}_n(P)$ is as in the theorem's statement. Thus,

$$\frac{(\mathbf{E}_{P(h)}|S_n^h|)^2}{2(\rho+1)\mathbf{E}_{P(h)}[V_n^h] + 2\mathbf{E}_{P(h)}|S_n^h|} \leq \mathrm{C}_n(P) \coloneqq \mathrm{KL}(P\|P_0) + \ln \frac{\phi_\rho(n)}{\delta}, \tag{31}$$

holds only if,

$$\frac{\mathbf{E}_{P(h)}[S_n^h]^2}{2(\rho+1)n\widehat{V}_n(P) + 2(\rho+1)\mathbf{E}_{P(h)}[S_n^h]^2/n + 2|\mathbf{E}_{P(h)}[S_n^h]|} \leq \mathrm{C}_n(P), \tag{32}$$

where we used the bias-variance decomposition in (30) together with the facts that $|\mathbf{E}_{P(h)}[S_n^h]| \leq \mathbf{E}_{P(h)}[|S_n^h|]$ (Jensen's inequality) and that the function $x \mapsto x^2/(x+v)$ is increasing on $\mathbb{R}_{\geq 0}$ for all $v > 0$. On the other hand, (32) is true for $P \in \mathcal{P}_n$, only if,

$$\left|\mathbf{E}_{P(h)}[S_i^h]\right| \leq \frac{2\mathrm{C}_n(P)/n + \sqrt{2(\rho+1)\widehat{V}_n(P) \cdot \mathrm{C}_n(P)/n}}{1 - 2(\rho+1)\mathrm{C}_n(P)/n}. \tag{33}$$

Thus, (31) holds only if (33) is true, and so we obtain the desired result by Proposition 5. $\qquad \square$

## C  Proofs of Monotonicity and Excess Risk Rates

To simplify notation in this section, we define

$$\widehat{L}_n(h) \coloneqq \frac{1}{n}\sum_{i=1}^n \ell(h, Z_i), \quad \widehat{L}(Q) \coloneqq \mathbf{E}_{Q(h)}[\widehat{L}_n(h)], \quad \text{for all } Q \in \triangle(\mathcal{H}).$$

We start by presenting a sequence of intermediate results needed in the proofs of Theorems 8 and 9.

### C.1  Intermediate Results

We now present a bound on the risk difference $L(Q) - L(Q')$, for any $Q, Q' \in \triangle(\mathcal{H})$, using our new time-uniform empirical Bernstein inequality in Theorem 6. For $\delta \in (0, 1)$, $\rho > 1$ and $k \geq 1$, we recall the definitions

$$\epsilon_k \coloneqq \frac{2\left(\mathrm{KL}(\mathrm{B}(Z_{1:k}) \times P_{k-1}\|P_0 \times P_0) + \ln \frac{\phi_\rho(k)}{\delta}\right)}{k \cdot (\rho+1)^{-1}}; \quad n_\delta \coloneqq \sup\left\{n : 8(\rho+1)\ln \frac{\phi_\rho(n)}{\delta} > n\right\}, \tag{34}$$

where $(P_k)$ are the outputs of Algorithm 1 and $\phi_\rho$ is as in Proposition 5.

**Lemma 24.** *Let $\rho > 0$, $P_0 \in \triangle(\mathcal{H})$, and $\mathcal{Q}_n$ be as in (8). Further, let $\delta \in (0,1)$ and $n_\delta$ as in (34). Then, under Assumption 1, we have, with probability at least $1 - \delta$, for all $n \geq n_\delta$ and $Q, Q' \in \mathcal{Q}_n$,*

$$L(Q) - L(Q') \leq \widehat{L}_n(Q) - \widehat{L}_n(Q') + \frac{\sqrt{\frac{\widehat{V}_n(Q,Q') \cdot \varepsilon_n(Q,Q')}{n}} + \frac{2\varepsilon_n(Q,Q')}{\rho+1}}{1 - \varepsilon_n(Q,Q')},$$

$$\text{where} \quad \varepsilon_k(Q,Q') \coloneqq \frac{2(\rho+1)(\mathrm{KL}(Q \times Q' \| P_0 \times P_0) + \ln \frac{\phi_\rho(k)}{\delta})}{k} \quad \text{and} \tag{35}$$

$$\widehat{V}_k(Q,Q') \coloneqq \frac{1}{k} \sum_{t=1}^{k} \mathbf{E}_{Q_k(h,h')}\left[(\ell(h,Z_t) - \ell(h',Z_t))^2\right] - \left(\frac{1}{k} \sum_{t=1}^{k} \mathbf{E}_{Q_k(h,h')}[\ell(h,Z_t) - \ell(h',Z_t)]\right)^2.$$

**Proof of Lemma 24.** The proof follows by our new time-uniform concentration inequality in Theorem 6 with the function $f : \mathcal{H}^2 \times \mathcal{Z} \to [0,1]$ defined by

$$f((h,h'),z) = (\ell(h,z) - \ell(h',z) + 1)/2.$$

Theorem 6 implies that, for any $\delta \in (0,1)$, with probability at least $1 - \delta$,

$$\mathbf{E}_{Q(h),Q'(h')}[L(h) - L(h') + 1]/2 \leq \frac{1}{n} \sum_{i=1}^{n} \mathbf{E}_{Q(h),Q'(h')}[f((h,h'),Z_i)] + \frac{\sqrt{\widehat{V}_n \varepsilon_n} + \frac{\varepsilon_n}{\rho+1}}{1 - \varepsilon_n}, \tag{36}$$

for all $n \geq n_\delta$ and $Q, Q' \in \mathcal{Q}_n$, where $\varepsilon_n = \varepsilon_n(Q,Q')$ and $\widehat{V}_n$ is given by:

$$\widehat{V}_n = \frac{1}{n} \sum_{t=1}^{n} \mathbf{E}_{Q(h),Q'(h')}\left[\left(f((h,h'),Z_t) - \frac{1}{n}\sum_{i=1}^{n} \mathbf{E}_{Q(\tilde{h}),Q'(\tilde{h}')}[f((\tilde{h},\tilde{h}'),Z_i)]\right)^2\right],$$

$$= \frac{1}{4n} \sum_{t=1}^{n} \mathbf{E}_{Q(h),Q'(h')}\left[\left(\ell(h,Z_t) - \ell(h',Z_t) - \frac{1}{n}\sum_{i=1}^{n} \mathbf{E}_{Q(\tilde{h}),Q'(\tilde{h}')}[\ell(\tilde{h},Z_i) - \ell(\tilde{h}',Z_i)]\right)^2\right],$$

$$= \frac{1}{4n} \sum_{t=1}^{n} \mathbf{E}_{Q(h),Q'(h')}\left[(\ell(h,Z_t) - \ell(h',Z_t))^2\right] - \left(\frac{1}{2n}\sum_{t=1}^{n} \mathbf{E}_{Q(h),Q'(h')}[\ell(h,Z_t) - \ell(h',Z_t)]\right)^2.$$

Plugging this into (36) and multiplying the resulting inequality by 2, leads to the desired inequality. $\qquad\square$

Lemma 24 leads to the following corollary that will be useful for our excess risk rates:

**Corollary 25.** *Let $\rho > 0$, $P_0 \in \triangle(\mathcal{H})$, and $\mathcal{Q}_n$ be as in (8). Under Assumption 1, we have for $\delta \in (0,1)$ and $n_\delta$ as in (34), with probability at least $1 - \delta$,*

$$L(Q) - L(Q') \leq \widehat{L}_n(Q) - \widehat{L}_n(Q') + 2\sqrt{\frac{\sum_{i=1}^{n} \mathbf{E}_{Q(h),Q'(h')}[(\ell(h,Z_i) - \ell(h',Z_i))^2] \cdot \varepsilon_n}{n}} + \frac{4\varepsilon_n}{\rho+1},$$

*for all $n \geq n_\delta$ and $Q, Q' \in \mathcal{Q}_n$, where $\varepsilon_k \coloneqq \frac{2(\rho+1)}{k}\left(\mathrm{KL}(Q \times Q' \| P_0 \times P_0) + \ln \frac{\phi_\rho(k)}{\delta}\right)$.*

**Proof of Corollary 25.** Let $\varepsilon_n(Q,Q')$ and $\widehat{V}_n(Q,Q')$ be as in Lemma 24. The corollary follows by Lemma 24 and the facts that $1 - \varepsilon_n(Q,Q') \geq 1/2$, for all $n \geq n_\delta$ and $Q, Q' \in \mathcal{Q}_n$; and

$$\widehat{V}_n(Q,Q') \leq \frac{1}{k} \sum_{t=1}^{k} \mathbf{E}_{Q_k(h,h')}\left[(\ell(h,Z_t) - \ell(h',Z_t))^2\right].$$

$\qquad\square$

The next lemma provides a way of bounding the square-root term in the previous corollary under the Bernstein condition (Definition 3):

**Lemma 26.** *Let $B > 1$ and $\beta \in [0,1]$, and suppose that the $(\beta, B)$-Bernstein condition holds. Further, let $\rho > 1$, $\delta \in (0,1)$, and $\varepsilon_k(Q, Q')$ be as in (35), for $Q, Q' \in \triangle(\mathcal{H})$. Then, under Assumptions 1 and 2, there exists a universal constant $C > 0$ s.t. with probability at least $1 - \delta$,*

$$\sqrt{\frac{\sum_{i=1}^n \mathbf{E}_{Q(h)}[(\ell(h, Z_i) - \ell(h_\star, Z_i))^2] \cdot \varepsilon_n(Q, Q')}{2^{-5}n}} \leq \frac{L(Q) - L(h_\star)}{2}$$
$$+ C \max_{\beta' \in \{\beta, 1\}} \varepsilon_n(Q, Q')^{\frac{1}{2-\beta'}}, \qquad (37)$$

*for all $n \geq 1$ and $Q, Q' \in \triangle(\mathcal{H})$, where $h_\star \in \arg\inf_{h \in \mathcal{H}} L(h)$.*

**Proof of Lemma 26.** Applying the fact that $\sqrt{xy} \leq (\nu x + y/\nu)/2$, for all $\nu > 0$, to the LHS of (37) with

$$\nu = \frac{\eta}{8}, \quad x = \frac{1}{n} \sum_{i=1}^n \mathbf{E}_{Q(h)}[(\ell(h, Z_i) - \ell(h_\star, Z_i))^2], \quad \text{and} \quad y = 2^5 \varepsilon_n(Q, Q'),$$

which leads to, for all $\eta > 0$, and $k = 2^5$,

$$r_n(Q) := \sqrt{\frac{k \sum_{i=1}^n \mathbf{E}_{Q(h)}[(\ell(h, Z_i) - \ell(h_\star, Z_i))^2] \cdot \varepsilon_n(Q, Q')}{n}},$$
$$\leq \frac{\eta}{16n} \sum_{i=1}^n \mathbf{E}_{Q(h)}[(\ell(h, Z_i) - \ell(h_\star, Z_i))^2] + \frac{4k\varepsilon_n(Q, Q')}{\eta}. \qquad (38)$$

Now, let $C_\beta := \left((1-\beta)^{1-\beta}\beta^\beta\right)^{\frac{\beta}{1-\beta}} + 3/2(2B)^{\frac{1}{1-\beta}}$. By combining (38) and Lemma 19, we get, for any $\delta \in (0,1)$ and $\eta \in [0, 1/2]$, with probability at least $1 - \delta$,

$$\forall Q \in \triangle(\mathcal{H}), \forall n \geq 1, \quad r_n(Q) \leq (L(Q) - L(h_\star))/2 + C_\beta \cdot \eta^{\frac{1}{1-\beta}}/4$$
$$+ \frac{\mathrm{KL}(Q \| P_0) + \ln \delta^{-1}}{2n\eta} + \frac{4k\varepsilon_n(Q, Q')}{\eta},$$
$$\leq (L(h) - L(h_\star))/2 + C_\beta \cdot \eta^{\frac{1}{1-\beta}}/4 + \frac{(4k + 1/4)\varepsilon_n(Q, Q')}{\eta}. \qquad (39)$$

Now, minimizing the RHS of (39) over $\eta \in (0, 1/2)$ and invoking Lemma 20, we get, for any $\delta \in (0,1)$, with probability at least $1 - \delta$,

$$\forall Q \in \triangle(Q), \forall n \geq 1, \quad r_n(Q) \leq \frac{L(Q) - L(h_\star)}{2} + +2(16k + 1/2)\varepsilon_n(Q, Q')$$
$$+ \frac{C_\beta \cdot (3 - 2\beta)}{4(1-\beta)} \left(\frac{4(1-\beta)(4k + 1/4)\varepsilon_n(Q, Q')}{C_\beta}\right)^{\frac{1}{2-\beta}},$$
$$\leq \frac{L(Q) - L(h_\star)}{2} + 2(4k + 1/4)\varepsilon_n(Q, Q')$$
$$+ \frac{C_\beta^{\frac{1-\beta}{2-\beta}} \cdot (3 - 2\beta)}{4(1-\beta)} \left(4(1-\beta)(4k + 1/4)\varepsilon_n(Q, Q')\right)^{\frac{1}{2-\beta}}. \qquad (40)$$

Combining (40) with the fact that $\beta \mapsto C_\beta^{\frac{1-\beta}{2-\beta}}$ is bounded in $[0,1)$, we get the desired result. $\qquad \square$

We now move on to the proofs of the main results of Section 4.

## C.2 Proofs of Theorems 8 and 9

Let $(\xi_k)$ and $n_\delta$ be as in (9) and (34), respectively. Further, it will be useful to define the event

$$\mathcal{E} := \left\{\forall n \geq n_\delta, \ L(\widetilde{P}_n) - L(P_{n-1}) \leq \widehat{L}_n(\widetilde{P}_n) - \widehat{L}_n(P_{n-1}) + \xi_n\right\}, \qquad (41)$$

where $\widetilde{P}_k := \mathsf{B}(Z_{1:k})$ and $(P_k)$ are as in Algorithm 1 with the choice of $(\xi_k)$ in (9). Observe that by Lemma 24, we have $\mathbf{P}[\mathcal{E}] \geq 1 - \delta$, under Assumptions 1 and 2. We begin by the proof of risk-monotonicity:

**Proof of Theorem 8.** Let $\Delta_n \coloneqq L(P_n) - L(P_{n-1})$. Using the definitions of $\mathcal{E}$ and $(\xi_k)$ as in (41) and (9), respectively, we have

$$\Delta_n = (L(\widetilde{P}_n) - L(P_{n-1})) \cdot \mathbb{I}\{P_n \not\equiv P_{n-1}\} + (L(P_n) - L(P_{n-1})) \cdot \mathbb{I}\{P_n \equiv P_{n-1}\},$$
$$= (L(\widetilde{P}_n) - L(P_{n-1})) \cdot \mathbb{I}\{P_n \not\equiv P_{n-1}\}. \tag{42}$$

Now, when $P_n \not\equiv P_{n-1}$, Line 2 of Algorithm 1 implies that

$$\widehat{L}_n(\widetilde{P}_n) \leq \widehat{L}_n(P_{n-1}) - \xi_n. \tag{43}$$

Using this and (42), we have that under the event $\mathcal{E}$,

$$\forall n \geq n_\delta, \quad L(\widetilde{P}_n) - L(P_{n-1}) \leq \widehat{L}_n(\widetilde{P}_n) - \widehat{L}_n(P_{n-1}) + \xi_n \leq 0.$$

This, combined with the fact that $\mathbf{P}[\mathcal{E}] \geq 1 - \delta$ (Lemma 24) completes the proof. $\qquad\square$

**Proof of Theorem 9.** Let $\widetilde{P}_k \coloneqq \mathsf{B}(Z_{1:k})$ and $(P_k)$ be as in Algorithm 1 with the choice of $(\xi_k)$ in (9). Further, we let $\epsilon_n$ be as in (9) and

$$\xi'_k \coloneqq 2\sqrt{\frac{\sum_{i=1}^n \mathbf{E}_{\widetilde{P}_k(h), P_{k-1}(h')}[(\ell(h, Z_i) - \ell(h', Z_i))^2] \cdot \epsilon_k}{k}} + \frac{4\epsilon_k}{\rho + 1}. \tag{44}$$

It will be convenient to also consider the events:

$$\mathcal{E} \coloneqq \left\{ \forall n \geq n_\delta, \ L(\widetilde{P}_n) - L(P_{n-1}) \leq \widehat{L}_n(\widetilde{P}_n) - \widehat{L}_n(P_{n-1}) + \xi'_n \right\},$$

$$\mathcal{E}' \coloneqq \left\{ \forall n \geq 1, \ Q, Q' \in \triangle(\mathcal{H}), \ \frac{\sqrt{\dfrac{\sum_{i=1}^n \mathbf{E}_{Q(h)}[(\ell(h, Z_i) - \ell(h_\star, Z_i))^2] \cdot \varepsilon_n(Q, Q')}{2^{-5}n}}}{\dfrac{L(Q) - L(h_\star)}{2} + C \cdot \left( \varepsilon_n(Q, Q')^{\frac{1}{2-\beta}} + \varepsilon_n(Q, Q') \right)} \leq \right\},$$

where $C$ and $\varepsilon_n(Q, Q')$ are as in Lemma 26. We note that by Corollary 25 and Lemma 26, we have

$$\mathbf{P}[\mathcal{E}] \wedge \mathbf{P}[\mathcal{E}'] \geq 1 - \delta. \tag{45}$$

For the rest of this proof, we will assume the event $\mathcal{E} \cap \mathcal{E}'$ holds, and let $n \geq n_\delta$ throughout. We consider two cases pertaining to the condition in Line 2 of Algorithm 1.

**Case 1.** Suppose that the condition in Line 2 of Algorithm 1 is satisfied for $k = n$. In this case, we have

$$L(P_n) - L(h_\star) = L(\widetilde{P}_n) - L(h_\star) \tag{46}$$

**Case 2.** Now suppose the condition in Line 2 does not hold for $k = n$. This means that $P_n \equiv P_{n-1}$, and so

$$\widehat{L}_n(P_n) - \widehat{L}_n(\widetilde{P}_n) \leq \xi_n \leq \xi'_n, \tag{47}$$

where the last inequality follows by the fact that $1 - \epsilon_n \geq 1/2$, for all $n \geq n_\delta$ under Assumption 2. Thus, by the assumption that $\mathcal{E}'$ is true, we have,

$$L(P_n) = L(\widetilde{P}_n) + (L(P_n) - L(\widetilde{P}_n)),$$
$$\leq L(\widetilde{P}_n) + \widehat{L}_n(P_n) - \widehat{L}_n(\widetilde{P}_n) + \xi'_n, \qquad (\mathcal{E} \text{ is true})$$
$$\leq L(\widetilde{P}_n) + 2\xi'_n, \qquad (\text{by (47)})$$
$$= L(\tilde{h}_n) + 4\sqrt{\frac{\sum_{i=1}^n \mathbf{E}_{\widetilde{P}_n(h), P_{n-1}(h)}[(\ell(h, Z_i) - \ell(h, Z_i))^2] \cdot \epsilon_n}{n}} + \frac{8\epsilon_n}{\rho + 1},$$
$$= L(\widetilde{P}_n) + 4\sqrt{\frac{\sum_{i=1}^n \mathbf{E}_{\widetilde{P}_n(h), P_n(h')}[(\ell(h, Z_i) - \ell(h', Z_i))^2] \cdot \epsilon_n}{n}} + \frac{8\epsilon_n}{\rho + 1}, \quad (P_n \equiv P_{n-1})$$
$$\leq L(\widetilde{P}_n) + 4\sqrt{\frac{2\sum_{i=1}^n \mathbf{E}_{\widetilde{P}_n(h)}[(\ell(h, Z_i) - \ell(h_\star, Z_i))^2] \cdot \epsilon_n}{n}} + \frac{8\epsilon_n}{\rho + 1}$$
$$+ 4\sqrt{\frac{2\sum_{i=1}^n \mathbf{E}_{P_n(h)}[(\ell(h, Z_i) - \ell(h_\star, Z_i))^2] \cdot \epsilon_n}{n}}, \tag{48}$$

where to obtain the last inequality, we used the fact that $(a-c)^2 \le 2(a-b)^2 + 2(b-c)^2$ and $\sqrt{a+b} \le \sqrt{a} + \sqrt{b}$ for all $a, b, c \in \mathbb{R}_{\ge 0}$. Now, by (48), the fact that $\mathcal{E}'$ holds, and Assumption 2 (which implies that $\epsilon_n^{\frac{1}{2-\beta}} \le O(\epsilon_n)$ for $n \ge n_\delta$), we have

$$L(P_n) - L(h_\star) \le L(\widetilde{P}_n) - L(h_\star) + \frac{L(\widetilde{P}_n) - L(h_\star)}{2} + \frac{L(P_n) - L(h_\star)}{2} + O\left(\epsilon_n\right)^{\frac{1}{2-\beta}},$$

which, after re-arranging, becomes

$$\frac{L(P_n) - L(h_\star)}{2} \le \frac{3(L(\widetilde{P}_n) - L(h_\star))}{2} + O\left(\epsilon_n\right)^{\frac{1}{2-\beta}}. \tag{49}$$

Multiplying on both sides by 2 and using (45) with a union bound leads to the desired result. $\qquad\square$

## C.3 Additional Results and Proofs

Using the lemmas in Section C.1, we derive the excess-risk rate of ERM under the Bernstein condition:

**Lemma 27.** *Let $B > 1$, $\beta \in [0, 1]$ and suppose that the $(\beta, B)$-Bernstein condition holds and $\mathcal{H}$ is finite. Further, let $\rho > 1$, $\delta \in (0, 1)$, and $n_\delta$ be as in (34). Then, under Assumptions 1 and 2, the ERM $\hat{h}_n \in \arg\min_{h \in \mathcal{H}} \frac{1}{n} \sum_{t=1}^{n} \ell(h, Z_t)$ satisfies, with probability at least $1 - \delta$,*

$$L(\hat{h}_n) - L(h_\star) \le O\left(\frac{\ln(\ln(n|\mathcal{H}|)/\delta)}{n}\right)^{\frac{1}{2-\beta}} + \frac{\ln(\ln(n|\mathcal{H}|)/\delta)}{n}, \tag{50}$$

*for all $n \ge n_\delta \vee (16(\rho+1) \ln |\mathcal{H}|)$.*

**Proof of Lemma 27.** Let $n_\delta$ be as in (34) and define

$$\epsilon_k := \frac{2(\rho+1)\left(2 \ln |\mathcal{H}| + \ln \frac{\phi_\rho(k)}{\delta}\right)}{k}, \quad \text{and} \quad \xi'_n := 2\sqrt{\frac{\sum_{i=1}^{n}(\ell(\hat{h}_n, Z_i) - \ell(h_\star, Z_i))^2 \cdot \epsilon_n}{n}} + \frac{4\epsilon_n}{\rho+1}.$$

Further, consider the events

$$\mathcal{E} := \left\{\forall n \ge n_\delta, \ L(\hat{h}_n) - L(h_\star) \le \widehat{L}_n(\hat{h}_n) - \widehat{L}_n(h_\star) + \xi'_n\right\},$$

$$\mathcal{E}' := \left\{\forall n \ge 1, \ \sqrt{\frac{\sum_{i=1}^{n}(\ell(\hat{h}_n, Z_i) - \ell(h_\star, Z_i))^2 \cdot \epsilon_n}{2^{-5}n}} \le \frac{L(\hat{h}_n) - L(h_\star)}{2} + C \cdot \left(\epsilon_n^{\frac{1}{2-\beta}} + \epsilon_n\right)\right\},$$

where $C$ is as in Lemma 26. By Corollary 25 and Lemma 26, instantiated with $P_0$ equal to the uniform prior over $\mathcal{H}$ and $Q$ [resp. $Q'$] equal to the Dirac at $\hat{h}_n$ [resp. $h_\star$], we have

$$\min(\mathbf{P}[\mathcal{E}], \mathbf{P}[\mathcal{E}']) \ge 1 - \delta. \tag{51}$$

For the rest of this proof, we will assume that the event $\mathcal{E} \cap \mathcal{E}'$ holds, and let $n \ge n_\delta$. By the assumption that $\mathcal{E}$ holds, we have

$$
\begin{aligned}
L(\hat{h}_n) &= L(h_\star) + (L(\hat{h}_n) - L(h_\star)), \\
&\le L(h_\star) + \widehat{L}_n(\hat{h}_n) - \widehat{L}_n(h_\star) + \xi'_n, &&(\mathcal{E} \text{ is true}) \\
&\le L(h_\star) + \xi'_n, &&(\hat{h}_n \text{ is the ERM}) \\
&= L(h_\star) + 2\sqrt{\frac{\sum_{i=1}^{n}(\ell(\tilde{h}_n, Z_i) - \ell(h_\star, Z_i))^2 \cdot \epsilon_n}{n}} + 4\epsilon_n. &&(52)
\end{aligned}
$$

Now by the assumption that $\mathcal{E}'$ holds, we can bound the middle term on the RHS of (52), leading to

$$L(\hat{h}_n) = L(h_\star) + \frac{L(\hat{h}_n) - L(h_\star)}{2} + O\left(\max_{\beta' \in \{1, \beta\}}\left(\frac{\ln(n|\mathcal{H}|/\delta)}{n}\right)^{\frac{1}{2-\beta'}}\right) + 4\epsilon_n,$$

$$= L(h_\star) + \frac{L(\hat{h}_n) - L(h_\star)}{2} + O\left(\frac{\ln(n|\mathcal{H}|/\delta)}{n}\right)^{\frac{1}{2-\beta}}, \tag{53}$$

for all $n \ge n_\delta \vee (16(\rho+1) \ln |\mathcal{H}|)$, where in the last inequality we used the definition of $\epsilon_n$. Combining (53) with (51), and applying a union bound, we obtain the desired result. $\qquad\square$

**Proof of Theorem 11.**    First, note that by linearity of the expectation it suffices to show that
$$\mathbf{E}\left[L(P_n) - L(P_{n-1})\right] \leq 0,$$
where the expectation is over the randomness of the samples $Z_{1:n}$. Moving forward, we let $\Delta_n \coloneqq L(P_n) - L(P_{n-1})$, and for $n \geq N$, define the event
$$\mathcal{E}_n \coloneqq \left\{ L(\widetilde{P}_n) - L(P_{n-1}) \leq \widehat{L}_n(\widetilde{P}_n) - \widehat{L}_n(P_{n-1}) + \xi'_n \right\}, \tag{54}$$
where $\widetilde{P}_k \coloneqq \mathsf{B}(Z_{1:k})$ and $(P_k)$ as in Algorithm 1 with the choice of $(\xi'_k)$ in the theorem's statement. Observe that by Lemma 24, we have $\mathbf{P}[\mathcal{E}_n] \geq 1 - 1/n^b$ for all $n \geq N$, under Assumptions 1 and 2.

Now, by the law of the total expectation, we have
$$\mathbf{E}[\Delta_n] = \mathbf{P}[\mathcal{E}_n] \cdot \mathbf{E}[\Delta_n \mid \mathcal{E}_n] + \mathbf{P}[\mathcal{E}_n^{\mathrm{c}}] \cdot \mathbf{E}[\Delta_n \mid \mathcal{E}_n^{\mathrm{c}}],$$
$$\leq \mathbf{P}[\mathcal{E}_n] \cdot \mathbf{E}[\Delta_n \mid \mathcal{E}_n] + 1/n^b.$$
where the last inequality follows by the fact that the loss $\ell$ takes values in $[0,1]$ and that $\mathbf{P}[\mathcal{E}_n^{\mathrm{c}}] \leq 1/n^b$. By applying the law of the total expectation again, we obtain
$$\mathbf{E}[\Delta_n] = \mathbf{P}[\{P_n \equiv P_{n-1}\} \cap \mathcal{E}_n] \cdot \mathbf{E}[\Delta_n \mid \{P_n \equiv P_{n-1}\} \cap \mathcal{E}_n]$$
$$+ \mathbf{P}[\{P_n \not\equiv P_{n-1}\} \cap \mathcal{E}_n] \cdot \mathbf{E}[\Delta_n \mid \{P_n \not\equiv P_{n-1}\} \cap \mathcal{E}_n] + 1/n^b,$$
$$\leq \mathbf{P}[\{P_n \not\equiv P_{n-1}\} \cap \mathcal{E}_n] \cdot \mathbf{E}[\Delta_n \mid \{P_n \not\equiv P_{n-1}\} \cap \mathcal{E}_n] + 1/n^b, \tag{55}$$
where the last inequality follows by the fact that if $P_n \equiv P_{n-1}$, then $\Delta_n = 0$. Now, if $P_n \not\equiv P_{n-1}$, then by Line 2 of Algorithm 1, we have
$$\widehat{L}_n(P_n) = \widehat{L}_n(\widetilde{P}_n) \leq \widehat{L}_n(P_{n-1}) - \xi'_n, \tag{56}$$
Under the event $\mathcal{E}_n$, we have
$$L(\widetilde{P}_n) - L(P_{n-1}) \leq \widehat{L}_n(\widetilde{P}_n) - \widehat{L}_n(P_{n-1}) + \xi'_n.$$
This, in combination with (56), implies that under the event $\mathcal{E}_n \cap \{P_n \not\equiv P_{n-1}\}$,
$$\Delta_n = L(\widetilde{P}_n) - L(P_{n-1}) \leq -\xi'_n + \xi'_n = 0.$$
As a result, we have
$$\mathbf{E}[\Delta_n \mid \{P_n \not\equiv P_{n-1}\} \cap \mathcal{E}_n] \leq 0. \tag{57}$$
Combining (55) and (57) yields the desired result.    □

**Proof of Proposition 10.** The risk monotonicity claim follows from Theorem 8, and the excess risk rate follows from Theorem 9 and Lemma 27.    □

# D   Risk Monotonicity without PAC-Bayes

In this section, we show how risk monotonicity can be achieved in the i.i.d. setting without Assumption 2. For this, we will use a concentration inequality due to [35] that has an empirical variance term under the square root just like ours in Theorem 6. To present this inequality, we first present some new notation. For any $Z_{1:n} \in \mathcal{Z}^n$, we let $\ell \circ \mathcal{H}(Z_{1:n}) \coloneqq (\ell(h, Z_1), \dots, \ell(h, Z_n))$. Further, for any subset $\mathcal{A} \subset \mathbb{R}^n$ and $\epsilon > 0$, we let $\mathcal{N}(\epsilon, \mathcal{A}, \|\cdot\|_\infty)$ be the cardinality of smallest subset $\mathcal{A}_0 \subseteq \mathcal{A}$ such that $\mathcal{A}$ is contained in the union of $\|\cdot\|_\infty$-balls of radii $\epsilon$ centered at points in $\mathcal{A}_0$. Finally, we consider the following complexity measure:
$$\mathcal{N}_\infty(\epsilon, \ell \circ \mathcal{H}, n) \coloneqq \sup_{Z_{1:n} \in \mathcal{Z}^n} \mathcal{N}(\epsilon, \ell \circ \mathcal{H}(Z_{1:n}), \|\cdot\|_\infty). \tag{58}$$

With this, we state the concentration inequality due to [35] that we will need:

**Theorem 28.** *Let $Z$ be a random variable with values in a set $\mathcal{Z}$ with distribution $\pi$, and let $\mathcal{H}$ be a set of hypotheses. Further, let $\delta \in (0,1)$, $n \geq 16$, and set*
$$\mathcal{M}(n) \coloneqq 10\mathcal{N}_\infty(1/n, \ell \circ \mathcal{H}, 2n).$$
*Then, with probability at least $1 - 2\delta$ in the random vector $Z_{1:n} \sim \pi^n$, we have*
$$\forall h \in \mathcal{H}, \quad \left| \mathbf{E}\left[\ell(h, Z)\right] - \frac{1}{n}\sum_{i=1}^n \ell(h, Z_i) \right| \leq \sqrt{\frac{18 V_n \ln(\mathcal{M}(n)/\delta)}{n}} + \frac{15\ln(\mathcal{M}(n)/\delta)}{n-1},$$
*where $V_n \coloneqq V_n(\ell \circ \mathcal{H}, Z_{1:n}) \coloneqq \frac{1}{n(n-1)} \sum_{1 \leq i < j \leq n} \left(\ell(h, Z_i) - \ell(h, Z_j)\right)^2$.*

---

**Algorithm 2** A Deterministic Risk Monotonic Algorithm Wrapper

---

**Require:** A base learning algorithm $\hat{h} : \bigcup_{i=1}^{\infty} \mathcal{Z}^i \to \mathcal{H}$.

Initial hypothesis $\hat{h}_0 \in \mathcal{H}$.

Samples $Z_1, \dots, Z_n$.

1: **for** $k = 1, \dots, n$ **do**

2:      Set $\widehat{V}_k \coloneqq \dfrac{1}{k(k-1)} \sum\limits_{1 \leq i < j \leq k} \left( \ell(h(Z_{1:k}), Z_i) - \ell(\hat{h}_{k-1}, Z_i) - \ell(h(Z_{1:k}), Z_j) + \ell(\hat{h}_{k-1}, Z_j) \right)^2$.

3:      Set $\xi_k = \sqrt{\dfrac{18\widehat{V}_k \ln(\mathcal{M}(k)/k)}{k}} + \dfrac{30 \ln(\mathcal{M}(k)/k)}{k-1}$.

4:      **if** $\dfrac{1}{k} \sum\limits_{i=1}^{k} \ell(\hat{h}(Z_{1:k}), Z_i)] - \dfrac{1}{k} \sum\limits_{i=1}^{k} \ell(\hat{h}_{k-1}, Z_i) \leq -\xi_k$ **then**

5:          Set $\hat{h}_k = \hat{h}(Z_{1:k})$.

6:      **else**

7:          Set $\hat{h}_k = \hat{h}_{k-1}$.

8: Return $\hat{h}_n$.

---

Using Theorem 28 and following the same steps in the proof of Theorem 11, it follows that Algorithm 2 is risk monotonic in expectation (up to an additive $2/k$ term) for all sample sizes. Furthermore, since the concentration inequality in Theorem 28 has an empirical variance term under the square-root (just like ours in Theorem 6), the risk decomposition in our Theorem 9 also holds for Algorithm 2, albeit with probability at least $1 - O(1/n)$ for sample size $n$.