# OpenReview forum: "Risk Monotonicity in Statistical Learning"
_NeurIPS.cc/2021/Conference — NeurIPS 2021 Oral_

### Official Review · Reviewer_kfNz · 2021-07-14

**Rating:** 9
**Confidence:** 4

**Summary:**

This work derives several novel concentration inequalities and uses them to derive the first risk monotonic algorithms for general statistical learning with bounded loss.

**Ethical Concerns:**

not that I can see

**Limitations And Societal Impact:**

not that I can see

**Main Review:**

General recommendation

This paper works on the important problem of risk-monotonic learning which has gotten much renewed interest since the discovery of the double descent phenenoma which is much more widely spread than previously throught. This paper, incredibly, manages to build a risk monotonic learning algorithm for the very general setting of bounded losses, for _any_ base learning algorithm. The algorithm is shown to be risk monotone in expectation and high probability, depending on the choice of the gaps. It remains very novel and surprising to me that it is at all possible to derive an algorithm which is risk monotonic in expectation for such a general setting. As such, I expect this to be a high impact paper on which future papers will build to further construct improved risk monotonic algorithms, which may open up a whole new field of study for machine learning algorithms and theory.

Sadly I am not an well-versed enough in concentration inequalities so I could not check the proofs in the limited time I have for reviewing this paper, I hope another reviewer can step in for this. If the math checks out, I definitely vote to accept this paper.  I did find a few minor issues and possibly misstakes, but I'm sure these issues can be resolved through the review process. See below.

Most important feedback

1) Assumption 2: can this assumption not be at odds with consistency? Or actually.... If we assume that the KL distance of an algorithm B is finite to the prior, can we not always adapt the algorithm B to satisfy this assumption? I mean, if the assumption is violated, we can just instead of using B's output, use the output of B in the previous round, until the condition is actually satisfied. So this seems to indicate that consistency will not be violated? I guess though that an essential condition is that the risk minimizer should be in an area of P_0 with > 0 probability? Maybe it would be good to clarify that explicitly.

2) Proposition 9: the point mass assumption seems incompatible to Assumption 2 to me, which is a condition of Theorem 8. So is there a misstake here?

3) Line 341: but for \beta = 0 (e.g. bounded losses), there is a cost to pay? Then I find in Line 173-174 the statement 'with essentially the same excess risk rate' slightly misleading. It only is shown that under the Bernstein condition beta > 0 the risk rate does not get worse significantly, right? Similairly for the statement in line 340 "risk monotonicity need not come at the price ....".

4) It strikes me that we need to resort to a randomized algorithm that fullfills Assumption 2. Can you provide some intuition why this was necessary? Furthermore, I do appreciate that the current algorithm works with non-i.i.d. data, but is the complicated new concentration inequality necessary? Why can we not use good old existing results? E.g. is Theorem 6 really necessary, or can I also drop in another inequality here?  If so, which would I choose? What makes it essential that we need this one? Is it that for this specific one we can analyze the algorithm under the Bernstein condition, but could we also obtain risk monotone algorithms with a simpler technique and or inequality? If you can come to a simpler version of the algorithm, or provide a recipe to come to that, that would be much appreciated :). But of course, if this is unfeasible, I would really include some more arguments to justify the current approach.

5) It is never reflected on what the intuition of \rho is. Do you have any idea what the consequence of this parameter is? How would you set it, or would you somehow tune it on the fly? I guess you may want to choose the largest or smallest value possible that still satisfies Assumption 2? Can you give some small explenation.

Small issues

6) Algorithm 1 is a bit misleading, since the sequence of gaps cannot be generated beforehand, as the gaps depend on the outputs of the base algorithm (e.g. on B(Z_{1:k}) and the P_k's). Would be good to squeeze it in between line 1 and 2: "compute \epsilon_k according to (eq. 9) for (\delta,n_\delta)-risk monotonicity, and to eq. X for risk-monotonicity for in expectation". Please also indicate the dependence of the gaps on the P_0, Q_k's, P_k's, P_{k-1}, rho, delta, ...). Would also be nice if it is indicated that rho, delta may be inputs to the Algorithm.

7) Line 364-366: this remark is a bit too much info in too little space, I could not understand it. Can this be explained in a bit more detail? Perhaps in the appendix. Why is convexity in the first argument enough for this construction and how would the construction go exactly? What kind of OCO algorithm would we need; e.g. one with sublinear regret is enough or we need something more advanced?

8) Line 193: it is never clarified what is meant by easier. Does it mean the gaps are bigger for the expectation case? Or is there somehow a cost in the rate of convergence?

9) Lines 298-300: I think this could definitely be non-trivial. Also, tuning the regularization or limiting the optimization may break guarantees of the base algorithm (e.g. a fast rate). So it seems there may be a trade-off still between learning speed and monotonicity....?

10) In the related work, you could consider
Viering, Tom, and Marco Loog. "The Shape of Learning Curves: a Review." arXiv preprint arXiv:2103.10948 (2021).
as it discusses even more causes of non-monotonic learning curves besides dipping and peaking.
Furthermore, you may be interested to know that peaking can be traced back historically already to 1989, see also
Loog, Marco, et al. "A brief prehistory of double descent." Proceedings of the National Academy of Sciences 117.20 (2020): 10625-10626.

11) Definition 2 seems a bit confusing; is this now a Theorem or Proposition or a definition? It seems like almost a theorem or proposition, but it cannot be correct. I guess it should be "Under Assumption 1 and Consistency of the Base Algorithm A is consistent" ?  But then probably it should also mention Assumption 2, and how the gaps are specified. Anyhow, it seems clear that if B is consistent, A must also be consistent, under Assumption a 1 & 2 and for both formula's of the gaps, correct?

**Time Spent Reviewing:**

3

---

> ### Author Response · Authors · 2021-08-10
> **We answer specific questions**
>
> Thank you for your detailed and valuable feedback. Here are answers to your questions:
>
> 1- Note that $(\mathcal{Q}_k)$ is a growing sequence of sets which cover the whole set of distributions $\Delta(\mathcal{H})$ in the limit $k \rightarrow \infty$. As a result, for large enough $k$, $(\mathcal{Q}_n), {n\geq k}$ will contain distributions with enough mass around the risk minimizer. Consistent algorithms will naturally converge to such distributions, and so they would likely be elements of $\mathcal{Q}_n$, for all $n\geq k$. As you suggest, for smaller sample sizes, we can use the output of B in the previous round until the condition is actually satisfied.
>
> 2- The point mass assumption of Proposition 9 is not incompatible with Assumption 2 since we have assumed $\mathcal{H}$ to be finite in the former. In this case, if the prior $P_0$ is the (discrete) uniform distribution over the finite $\mathcal{H}$, the KL-distance of $B(Z_{1:k})$ (a point mass in the setting of Prop 9) to the prior is just $\ln |\mathcal{H}|$. Thus, Assumption 2 would technically hold for all $k\geq 16 (\rho+1) \ln |\mathcal{H}|$ (we will clarify this point). We also realized that we did not explicitly say that the prior distribution is uniform in the statement of Proposition 9, which may have led to your question in the first place. We will fix this.
>
> 3- When the Bernstein condition only holds for $\beta=0$ and not for $\beta>0$, the best excess risk rate of any algorithm B under no additional assumptions on the problem is of order $\sqrt{\text{COMP}(\mathcal{H})/n}$, where $\text{COMP}({\mathcal{H}})$ is some measure of complexity of the set $\mathcal{H}$ (that may also be algorithm-dependent such as the KL-distance). On the other hand (as we touch on at the bottom of page 8), the additional price we pay in our excess risk rate is $(\epsilon_n)^{1/(2-\beta)}\stackrel{\beta=0}{=} \sqrt{\epsilon_n} \leq O(\sqrt{ \max_{k\leq n}KL(B(Z_{1:k})|| P_0) /n})$, which should be of the same order as $\sqrt{
> \text{COMP}(\mathcal{H})/n}$ even if $\text{COMP}(\mathcal{H})$ is a measure of complexity other than the KL-distance (KL-based generalization bounds tend to be tighter than others). An analogous story continues to hold for $\beta>0$, and this is why we claimed that there is virtually no price to pay when it comes to excess-risk rates. We will make our reasoning clearer in the final version.
>
> 4- Randomization is not required to achieve consistency and risk monotonicity. In fact, if one specifies the gap sequence using uniform convergence bounds or other types of bounds that allow for deterministic estimators (instead of PAC-Bayes), then one can show consistency and risk-monotonicity in the same way as we do in the proof of Theorem 7. The advantage that we get from using our concentration inequality is that we are able to achieve fast rates under the Bernstein condition. This is enabled by the presence of an empirical variance term in the bound. We are unaware of a concentration bound for deterministic estimators that has a similar dependence on the empirical variance (except perhaps for https://arxiv.org/pdf/0907.3740.pdf, but these would have a loose complexity term in addition to not holding for non-i.i.d. processes). What is more, the condition of Assumption 2 may be disposed of while maintaining fast rates under the Bernstein condition. The only reason we need Assumption 2 is to ensure that the denominator in our gap sequence remains bounded away from zero. Instead, one could use the PAC-Bayesian empirical Bernstein bound (https://papers.nips.cc/paper/2013/file/a97da629b098b75c294dffdc3e463904-Paper.pdf), which does not have any problematic denominators. Since this generalization bound also has the empirical variance under the main square-root term, it would be possible to show risk-monotonicity with fast rates under the Bernstein condition. The reason we chose to use Theorem 6 instead (at the cost of imposing Assumption 2) is to allow us to show risk monotonicity even for the non-i.i.d. processes of Assumption 1. It also enables us to have near-optimal dependence in the sample size $n$ in the final rates; we get optimal $\ln \ln n$ factors instead of $\ln n$ (please see point 2 in response to 99EG). In summary, it is possible to use other concentration bounds other than the one in Theorem 6. This can remove the requirement of Assumption 2 and allow for deterministic estimators. However, The price to pay for this may be slower excess-risk rates and no risk-monotonicity for non-i.i.d. processes. We will add a discussion on this point.
>
> 5- $\rho>1$ is a hyper-parameter of our concentration bound. Ideally, one would set $\rho =1$, but this would make the $\log$ terms in the bound explode. So in practice, I would expect a value of $\rho\in(1,2)$ to be optimal. Alternatively, one can even tune the parameter after seeing the data by applying a simple union bound over a finite grid of $\rho$'s.
>
> 6 - Good point; we will fix this.
>
> 7- We will add more details in the appendix on the convex case (including a pseudo-code). In short, yes, any algorithm with sublinear regret is enough.
>
> 8- Indeed, easier in the sense that the gap sequence is larger for the expected case as soon as $n\geq 1/\delta$.  In retrospect, 'easier' may not be the best word to use. We will address this.
>
> 9- As long as the algorithm remains consistent after turning the regulariser/covariance, our results on consistency and risk-monotonicity of Algorithm 1 would still hold. But it is true that the rate of convergence may worsen. We leave such investigations for future works.
>
> 10- Thank you for pointing out the missing related works. We will cite the works appropriately.
>
> 11- "Anyhow, it seems clear that if B is consistent, A must also be consistent, under Assumption a 1 & 2 and for both formula's of the gaps, correct?"
> If by A you mean Algorithm 1, then your statement is correct. In Definition 2, A is not referring to Algorithm 1 necessarily. It is defining what it means for an algorithm A to be consistent under the loss process defined in Assumption 1. We will rephrase the definition to make this clearer.

---

> > ### Comment · Reviewer_kfNz · 2021-08-27
> > **Thanks for all clarifications!**
> >
> > I must say I am already completely satisfied. But just a few little follow up questions on the above out of curiosity:
> >
> > 1. Is there any example you can give of a learning algorithm that would satisfy such an assumption in practice?
> > 4. I would be very interested in a minimum viable algorithm / proof (e.g. the 'simplest' possible). But I can understand it will be a lot of work to work it out and may not have time / space for it. I could suggest to put this in the Appendix. I think it would make the work even more accessible to a larger audience - because now the algorithm and the theoretical results have many parts. Anyhow, it would be much appreciated if the steps towards achieving this are discussed somewhere.
> > 5. I am still left wondering: what is the intuition? For example; what I would be looking for: if \rho is smaller, the algorithm is more conservative? Or the like? I see the usual trick with the union bound; I think for the high probability results this works and we get an additional term because of this. But would this also work for the expectation you think?
> >
> > Thanks!

---

> > > ### Author Response · Authors · 2021-08-27
> > > **Thanks for the feedback**
> > >
> > > We appreciate your positive feedback.
> > >
> > > 1- For most models and `well behaved' algorithms---those that are consistent and sufficiently stable---Assumption 2 is typically satisfied. This is because consistent and stable algorithms have a $KL$-divergence, $KL(P_n\|P_0)$, that grows sublinearly in the number of samples $n$. An example of a not-so-well-behaved algorithm is one that outputs some distribution $P_n$ centred around the last iterate of Gradient Descent (GD) used to train a very large Neural Network. In this case, the last iterate $\hat{w}$ of GD can be very unstable and have a large norm, which would in turn make the KL-divergence large. To see this, suppose that $P_n\equiv \mathcal{N}(\hat w, \sigma^2 I)$ and $P_0\equiv \mathcal{N}(0, \sigma^2 I)$, for some $\sigma>0$, then $KL(P_n\|P_0) = ||\hat w||^2/(2\sigma^2)$, and so the larger the norm of the last iterate $\hat w$, the larger the $KL$. It is possible to apply some heuristics to prevent $||\hat w||$ from being too large, by for example, adding a regularization term of the form $\lambda ||w||^2$ to the objective, and tune $\lambda>0$ until $||\hat w||$ is small enough for Assumption 2 to hold (one can always pick $\lambda>0$ large enough for this purpose). However, choosing $\lambda>0$ too larger may hurt the performance of the algorithm (e.g. in terms of the excess risk), and so one needs to pick the smallest $\lambda$ for which the assumption holds.
> > >
> > > Alternatively (as we explained in point 4 of our initial response), Assumption 2 can be circumvented using other generalization bounds to specify the gap sequence in our wrapper algorithm. Using the bound in Theorem 6 of Maurer and Pontil (2009) https://arxiv.org/pdf/0907.3740.pdf, for example, will further allow deterministic hypotheses. The price to pay for this simplification is that our wrapper algorithm will no longer be risk monotonic for loss processes satisfying our Assumption 1 (it will be risk-monotonic only for i.i.d. data), and its excess-risk will be larger than what you would get using our concentration bound (the bound of Mauer and Pontil (2009) is based on a uniform convergence argument, which typically leads to looser bounds compared to PAC-Bayes).
> > >
> > > 2- We are happy to include a section in the appendix where we would present the simplest viable algorithm without Assumption 2 or the need to randomize (i.e. no PAC-Bayes). This is not going to be much work. Thank you for your feedback on this.
> > >
> > > 3- The rule of thumb is that we want to choose $\rho$ to minimize our concentration bound (a smaller $\rho$ is not always better as the bound would explode for $\rho \downarrow 1$). The algorithm will be less conservative when picking a $\rho$ that minimizes the concentration bound (and, in turn, the gap sequence). Doing a union bound over $\rho$ will still work in expectation. The simplest way to do this is to think of $\rho$ as an extra parameter that need be learned. That is, you may consider the augmented hypotheses space $\mathcal{H}' = \mathcal{H}\times \mathcal{G}$ where $\mathcal{G}$ is a grip over possible values of $\rho$. This will allow you to pick $\rho$ after seeing the data, and all the results of our paper will hold if all instances of $KL(P_n|P_0)$ in our rates and elsewhere are replaced by $KL(P_n|P_0) + \ln |\mathcal{G}|$.

---

> > ### Public Comment · ~Kweku_Abraham1 · 2022-02-23
> > **Point 7**
> >
> > Is the version with details in the appendix on the convex case available? I couldn’t see anything in the latest version I could find.

---

### Official Review · Reviewer_mLBe · 2021-07-14

**Rating:** 7
**Confidence:** 3

**Summary:**

Risk monotonicity has raised many research interest since the pioneer work of Belkin, Mikhail, et al., as this contradicts the classical understanding that the risk decreases as we see more samples. The non-monotonic behavior of the risk curve stress that we lack a clear understanding of generalization. As a result, this paper aims to addressing the problem of how to obtain an algorithm that returns an estimator with monotonic risk behavior.

In addressing the above mentioned question, the authors make the following contributions:

The authors first derive a new concentration inequality for Martingale Difference Sequences. The bound drops the traditional assumption of i.i.d data and is time-uniform. In addition, the bound improves over the standard concentration inequalities by a factor of $\log n$.

On top of the concentration inequality bound, the authors also propose a "wrapper" algorithm which returns a consistent estimator that is proved to demonstrate monotonic risk curve. In addition, the algorithm does not hurt the excess risk convergence rate.

**Main Review:**

Clarity: The paper is clearly structured and well-written, though the authors are encouraged to carefully proof-read the manuscript to avoid some typos.

Quality \& Significance: I am not an expert on PAC-Bayesian analysis. However, as mentioned before, the non-monotonic risk behavior is a very important research question in the machine learning community and many papers have been published in understanding this phenomenon. Hence, this question that the authors study in this paper is of significant importance.

I do have some major concerns for the proposed algorithm. My first question is that, if I understand correctly, the proposed algorithm trades the monotonic risk curve with the computation time? If this is the case, then what exactly is the trade-off between the computation time and the monotonic risk behavior?

My second question is that can the authors design some simple numerical experiments to explore the efficacy of the proposed algorithm. In particular, it would be more convincing to show experimentally that the algorithm indeed return an estimator with monotonic risk behavior.

**Time Spent Reviewing:**

15

---

> ### Author Response · Authors · 2021-08-10
> **We answer specific questions**
>
> Thank you for the extensive time you took to carefully review our paper.
>
> The primary goal of this paper was to answer the fundamental question around the existence of a consistent, risk-monotonic algorithm in the general statistical learning setting. We answer this in the affirmative and further show that there is virtually no cost for achieving this when it comes to excess-risk rates. We believe this is an important milestone in the search for risk-monotonic algorithms. We note that the results just mentioned are supported by mathematical proof and do not necessarily require experimental validation.
>
> On the other hand, though the proposed algorithm is simple, it is not necessarily the most practical one in its current form. As can be seen in Algorithm 1, for a sample of size $n$, the base algorithm B need be called $n$ times (instead of only once if we do not care about risk-monotonicity) to return the final distribution over hypotheses. As a result, the proposed algorithm may be $n$ times slower than the base algorithm B itself if no additional structure of the problem is leveraged (such as convexity of the loss---see paragraph starting at line 364). In the special case where B$(Z_{1:k})$, $k\geq 1$, is a centred distribution around the ERM for samples $(Z_{1:k})$, it may be possible to efficiently evaluate the empirical risk w.r.t. the distributions $B(Z_{1:1}), \dots, B(Z_{1:n})$ (which is needed in our algorithm) using incremental updates, without incurring a significant computational overhead compared to evaluating the empirical risk only w.r.t. $B(Z_{1:n})$. In future work, we hope to focus more on such practical considerations, where we would also aim to test the empirical performance of the algorithms.

---

> > ### Comment · Reviewer_mLBe · 2021-08-29
> > **Thanks for the response**
> >
> > Thank you very much for your reply. I think the your response addresses my concern. I am happy to accept this paper.

---

### Official Review · Reviewer_99EG · 2021-07-17

**Rating:** 7
**Confidence:** 4

**Summary:**

The paper provides a wrapper algorithm that, for bounded loss functions, operates on any base learning algorithm B and outputs a learning algorithm whose risk decreases monotonically with access to more samples (a very desirable property in practice). The wrapper function is based on a new concentration inequality for martingale difference sequences which is derived using the regret bounds for a particular parameter free online learning algorithm.



**Limitations And Societal Impact:**

No social negative impact.

**Main Review:**

I think the paper is fairly complete in terms of the results, and provides sufficient theoretical results and new proof ideas. I recommend it for acceptance.  I have a few questions:

* Can you provide some examples where Assumption 1 holds but data is not i.i.d. ? Typically, we would like the behavior of the learning algorithm to be invariant to the order in which the samples are provided. Thus, we would like the property to hold independent of the order of samples. What kind of learning setting do you hope to use your monotonic algorithm where Assumption 1 holds for data is not i.i.d.?

* Can you provide some intuition on why we could not use empirical Bernstein inequalities for setting the value of \zeta_k ? Why is it natural to look at the problem of risk monotonicity in the PAC-Bayes setting ?

*  The results provided in the paper hold for all N >= N_\Delta. Is it possible to show that for every N_\Delta, there exists some distribution for which learning monotonically is impossible when N < N_\Delta?

* (Minor) In lines 128-129, the authors claim that the results of [49] could not be extended to hold in expectation. Can the authors discuss why ?

The paper is well written otherwise. I skimmed through the proofs and they seem to be correct.
The proof ideas in Appendix B are quite interesting!

Minor comments:
* References [22] and [23] are identical.
* Lines 36-40 regarding discussion of connections to neural networks seem to be opposite to the empirical observations.

**Time Spent Reviewing:**

10+ hours

---

> ### Author Response · Authors · 2021-08-10
> **We answer specific questions**
>
> Thank you for your review. Here are answers to your specific question:
>
> - We introduced Assumption 1 because we wanted to study the most general setting under which risk-monotonicity still makes sense. If Assumption 1 is not satisfied, then for any given hypothesis $h$ the expected risk $\mathbb{E}[\ell(h, Z_t)]$ may be different for every $t$. This means that requiring monotonicity and consistency at the same time may be conflicting since $L(h_{t,\star})=\mathbb{E}[\ell(h_{t,\star},Z_t)]$, where $h_{t,\star}$ is a minimizer of $h \mapsto \mathbb{E}[\ell(h,Z_t)]$, can vary, and so any `consistent' algorithm that attempts to track the optimal risk $L(h_{t,\star})$ may be forced to be non-monotonic. We believe that the non-i.i.d. process of Assumption 1 may still be relevant, not just in theory, but also in practice. However, we did not investigate which particular applications it would apply to. Nevertheless, here is a toy example where the non-i.i.d. process of Assumption 1 holds:
> Consider a regression setting where data consists of pairs $(X_t,y_t)$. Now suppose that each feature $X_t$ is of the form $X_t=(S_t,R_t)$, where $R_t$ is an Orthogonal matrix than can be chosen arbitrarily by an adversary and $S_t = R_t \tilde X_t$ with $(\tilde X_1,y_1),(\tilde X_2,y_2),\dots$ being i.i.d. Finally, let the loss be $\ell(h,Z_t) = (y_t - S_t^\top R_t h)^2$. Note that in this case, $\mathbb{E}[\ell(h,Z_t)]= \mathbb{E}[(y_t - \tilde X_t^\top h)^2]$ is fixed for all $t$ since $(\tilde X_1,y_1),(\tilde X_2,y_2),\dots$ are i.i.d., and so Assumption 1 holds for the loss process $(\ell(h,Z_t))$. Yet, this loss process is not necessarily i.i.d., since we allowed $R_t$ to be chosen arbitrarily (it could depend on previous samples, for example).
>
> - The empirical Bernstein inequality, together with a union bound over sample sizes, can be used to set the sequence of gaps $(\xi_k)$. Under an i.i.d. assumption, this would still lead to a risk-monotonic algorithm with a fast rate under the Bernstein condition with $\beta>0$. As we explain in the paragraph starting at line 268, our concentration bound allows us to achieve this for non-i.i.d. processes satisfying Assumption 1. Furthermore, our bound does not have suboptimal $\ln n$ factors. These would otherwise be present if one applies a concentration inequality such as Bernstein with a union bound over sample sizes. Instead, we have the optimal $\ln \ln n$ (optimal according to the law of iterated logarithms). Regarding why it matters to consider the PAC-Bayes setting, please see the answer to point 4 of Reviewer kfNz.
>
> - It is technically possible to achieve monotonicity for $n <n_\delta$ by just outputting a fixed hypothesis that does not depend on the data for all sample sizes $n<n_\delta$ (we touch on this in the case of risk-monotonicity in expectation at the end of Section 4). The risk of the corresponding algorithm will then be constant, thus risk-monotonic, for all $n<n_\delta$. We note that $n_\delta$ is relatively small; of order $O(\ln (1/\delta))$ (see the last paragraph of Section 2), and so it may not matter what happens at such low sample sizes.
>
> - The approach taken by Viering et al. in https://arxiv.org/pdf/1911.11030.pdf is in a way similar to ours but is based on selecting hypotheses based on their performance on a validation set. In our case, we can compare hypotheses (or distributions over hypotheses) on the training samples directly, and we pick the best ones using confidence bounds. Using this approach, we can get risk-monotonicity in probability and in expectation by simply modifying the sequence of gaps $(\xi_k)$ (please see paragraph starting at line 342). It is not clear to us how the approach of Viering et al. can be modified to achieve a monotonic guarantee in exception (or even in probability, for losses other than the 0-1).

---

> > ### Comment · Reviewer_99EG · 2021-09-13
> > **Post rebuttal!**
> >
> > Thank you for your response. I have read other reviews and the author feedback and would like to keep my review unchanged.  I continue to support acceptance for this paper!

---

### Official Review · Reviewer_qjbn · 2021-07-19

**Rating:** 6
**Confidence:** 3

**Summary:**

The paper studies the problem of risk monotonicity, that is, the property that the population loss decreases monotonically with increasing data points. The paper proposes an algorithm that converts any consistent base algorithm into a risk-monotonic algorithm while maintaining consistency under two assumptions: (1) an assumption on the generation of samples which is weaker than i.i.d. (2) an assumption on the base algorithm requiring its posterior to not have KL divergence to a fixed prior growing super-linearly in the sample size.

The approach presented in the paper is pretty straightforward: the base algorithm is run for $k=1, \ldots, n$ times and iteratively the current hypothesis is updated to be the base learner's output only if the risk reduces from the previous step, else it remains unchanged. Finally the hypothesis at step $n$ is returned. The main challenge here is to be able to confidently account for this change using the samples. For this, the authors derive a new concentration inequality for this using an online algorithm's guarantees.



**Limitations And Societal Impact:**

The authors should add more discussions about the practicality of the algorithm and its limitations in practice.

**Main Review:**

Developing risk monotonic algorithms has gained interest in the last few years due to the 'double descent' phenomenon observed in deep learning making the topic of the paper is pretty relevant to the community. I think the paper takes a good step in this direction. There are some new ideas in the paper though the algorithm itself is pretty straightforward. My main concerns are the practicality of the algorithm and the validity of the assumptions in practice. The paper would greatly benefit from an experimental evaluation to support the approach on some real-world problems.

Comments/Questions/Typos:
1. Typos:
	Line 2: populating -> population
	Line 175: continues -> continuous
2. Definition 2, 'converges in probability' is not clear.
3. Why is $\rho$ needed for the theorem statements? Can you not replace it by a constant?
4. Assuming access to the distribution and being able to calculate the expectation for the algorithm $B$ efficiently requires further discussion. For a randomized algorithm B, you would not expect to have access to the distribution over hypothesis, so you'll have to estimate that and run B another $\log(n/\delta)$ times?
5. Since you use the FreeGrad guarantee on a scalar, are there more direct ways to prove the desired inequaity. It seems to resemble a self-normalizing Martingale bounds.
6. The proofs using the ESI notation require more description. It would be good to add what basic operations hold under this operator and further eleaborate on the proofs.

**Time Spent Reviewing:**

6

---

> ### Author Response · Authors · 2021-08-10
> **We comment on the validity of the assumptions and answer specific questions**
>
> Thank you for the time you took to review our paper.
>
> Regarding the validity of the assumptions: the process assumption (Assumption 1) we make is weaker than the standard i.i.d. one. The base algorithm restriction (Assumption 2) can be easily removed if one does not care about non-i.i.d. processes by using a generalization bound based on the standard Bernstein concentration inequality (this would also enable fast rates). We make Assumption 2 only to bound the denominator in our concentration (Theorem 6) away from zero, which is not necessary for other concentration bounds. Please see reply to point 4 of Review kfNz for more details on this. Also, see the answer to Reviewer mLBe regarding experiments.
>
> Here are answers to your specific questions:
> - $\rho>1$ is a hyper-parameter of our concentration bound. Ideally, one would set $\rho =1$, but this would make the $\log$ terms in the bound explode. So in practice, we would expect a value of $\rho\in(1,2)$ to be optimal. Alternatively, one can even tune the parameter after seeing the data by applying a simple union bound over a finite grid of $\rho$'s.
> -  As we discuss between lines 298 and 300, the output distribution of Algorithm B is typically specified as a Gaussian around the ERM (or the final iterate of SGD). So in a practical setting, we would know the distribution. But, you are right; calculating expectations would require some form of estimation. This is typically done via Monte Carlo sampling. Alternatively, one may avoid expectations altogether using recent derandomization techniques, see e.g. https://arxiv.org/pdf/2102.08649.pdf. We will add a discussion on this point.
> - We suspect there may be other ways to obtain a similar concentration inequality to ours without the use of online learning. For example, the unexpected Bernstein inequality in Lemma 13 in https://papers.nips.cc/paper/2019/file/3dea6b598a16b334a53145e78701fa87-Paper.pdf, which can be seen as an empirical version of the classical Bernstein concentration inequality, may be a good starting point for deriving a similar bound to ours. The techniques used in e.g. https://arxiv.org/pdf/1808.03204.pdf can also be used for this purpose. However, we are unaware of a method that gets the same or better constants than ours in the final bound. By going through FreeGrad's guarantee, we were essentially able to build the `tightest' non-negative supermartingale from which we derived the concentration inequality.
>  - We will address the issues you raised in your second and sixth points appropriately.

---

> > ### Comment · Reviewer_qjbn · 2021-08-19
> > **Thanks for the response**
> >
> > Thank you for responding to my questions/comments. Thanks for clarifying the assumptions. My concern with practicality still remains. It would be good to add more discussion about the necessity of the computational overhead of running $B$ repeatedly.
> >
> > I still lean towards accepting the paper and will maintain my score.

---

> > > ### Author Response · Authors · 2021-08-20
> > > **We appreciate your feedback**
> > >
> > > We appreciate your feedback. We will expand more on the practicality/computational issues in the discussion section.

---

### Author Response · Authors · 2021-08-10
**We thank the reviewers for their time and valuable feedback**

We thank the reviewers for their time and valuable feedback. We hope we addressed all questions and concerns in our responses below.

---

### Decision · Program_Chairs · 2021-09-27

**Decision:**

Accept (Oral)

**Comment:**

This work makes a novel, surprising, and significant contribution to the area of risk monotonicity, a topic of great interest to the machine learning community. A key consequence of the authors results are a positive resolution of one of the open problems in Viering et al.'s COLT 2019 open problem paper, which is the question of whether it is possible to avoid non-monotonicity of the risk. The authors manage to achieve this in quite a general setting via a wrapper procedure which converts any base algorithm to a new algorithm that achieves risk monotonicity both in expectation and in high probability. That this would be possible at all in expectation was considered to be very surprising. The key to the authors’ approach is a novel concentration inequality for martingale difference sequences which is derived using a guarantee for the recent FreeGrad algorithm; the concentration inequality result itself may be of independent interest and could have impact in the future. In addition, I believe the risk monotonicity results themselves in this paper also may have great impact.

I also want to summarize some of the weaker points of the paper, as well as my and the reviewers' takes from the discussion:
- One point was raised about the validity of the assumptions, and the authors gave a convincing response on how Assumption 1 is quite weak and how Assumption 2 can be removed when one operates in the oft-studied i.i.d. setting.
- Multiple reviewers asked about practicality of the approach; given the theoretical progress made in this work, I myself view practicality as secondary, but even so, I encourage the authors to further discuss practicality/computational issues in the discussion section of their paper, as they have suggested they will do.
- A couple of reviewers also mentioned doing experimental validation, but I find that the theoretical contribution stands regardless of experiments, and if extra space was there, I imagine such extra space could instead be used to get into the ("quite interesting", as one reviewer put it) proof of the novel concentration inequality via FreeGrad.
- Finally, the question of whether randomization is needed came up, and whether we really need the novel concentration inequality. On this point, I found that the authors gave a convincing answer for why they use a novel concentration inequality (avoiding $\log n$, achieving fast rates when possible, etc.), and I do hope they will follow through with the showing a simple example in the appendix where there is no Assumption 2 and no randomization (no need for PAC-Bayes).

I have one comment regarding terminology:
- I disagree that the Bernstein condition with $\beta = 1$ is the Massart noise condition. The Massart noise condition is the best case of the Tsybakov margin condition (the best case), and the latter requires realizability (the Bayes optimal classifier must belong to the class $\mathcal{H}$); moreover, I am almost certain that the name "Massart noise condition" is exclusively used in the context of 0-1 loss. The authors may want to do a quick literature review to get the terminology right.

In conclusion, this is a solid paper that I expect will have high impact both from resolving an open problem and from the novel concentration inequality. This paper should definitely appear at NeurIPS.